# Simultaneous embedding of multiple attractor manifolds in a recurrent neural network using constrained gradient optimization

**Haggai Agmon**
The Hebrew University of Jerusalem, Israel
and Stanford University, USA
haggai.agmon@mail.huji.ac.il

**Yoram Burak**
The Hebrew University of Jerusalem, Israel
yoram.burak@elsc.huji.ac.il

## Abstract

The storage of continuous variables in working memory is hypothesized to be sustained in the brain by the dynamics of recurrent neural networks (RNNs) whose steady states form continuous manifolds. In some cases, it is thought that the synaptic connectivity supports multiple attractor manifolds, each mapped to a different context or task. For example, in hippocampal area CA3, positions in distinct environments are represented by distinct sets of population activity patterns, each forming a continuum. It has been argued that the embedding of multiple continuous attractors in a single RNN inevitably causes detrimental interference: quenched noise in the synaptic connectivity disrupts the continuity of each attractor, replacing it by a discrete set of steady states that can be conceptualized as lying on local minima of an abstract energy landscape. Consequently, population activity patterns exhibit systematic drifts towards one of these discrete minima, thereby degrading the stored memory over time. Here we show that it is possible to dramatically attenuate these detrimental interference effects by adjusting the synaptic weights. Synaptic weight adjustments are derived from a loss function that quantifies the roughness of the energy landscape along each of the embedded attractor manifolds. By minimizing this loss function, the stability of states can be dramatically improved, without compromising the capacity.

## Introduction

In the brain, recurrent neural networks (RNNs) involved in working memory tasks are thought to be organized such that their dynamics exhibit multiple attractor states, enabling the maintenance of persistent neural activity even in the absence of external stimuli. In tasks that require tracking of a continuous external variable, these attractors are thought to form a continuous manifold of neural activity patterns. In some well studied brain circuits in flies and mammals, neural activity patterns have been shown to robustly and persistently reside along a single, low-dimensional manifold, even when the neural activity is dissociated from external inputs to the network [2, 39, 21, 35, 9, 15]. In other brain regions, however, the same neural circuitry is thought to support multiple low-dimensional manifolds such that activity lies on one of these manifolds at any given time, depending on the context or task. The prefrontal cortex, for example, is crucial for multiple working memory and evidence accumulation tasks [14, 34, 46, 47], and it is thus thought that the synaptic connectivity in this brain region supports multiple attractor manifolds. Naively, however, embedding multiple attractor manifolds in a single RNN inevitably causes interference that degrades the network performance.

To be specific, we focus here on attractor models of spatial representation by hippocampal place cells [28]. The synaptic connectivity in area CA3 is thought to support auto-associative dynamics, and place cells in this area are often modeled as participating in a continuous attractor network

37th Conference on Neural Information Processing Systems (NeurIPS 2023).

(CAN) [7, 41, 40, 31, 49, 36, 11, 13, 17, 8]. Typically, neurons are first arranged on an abstract neural sheet which is mapped to their preferred firing location in the environment. The connectivity between any two neurons is then assumed to depend on their distance on the neural sheet, with excitatory synapses between nearby neurons and effectively inhibitory synaptic connections between far away neurons. This architecture constrains the population activity dynamics to express a localized self-sustaining activity pattern, or 'bump', which can be centered anywhere along the neural sheet. Thus, the place cell network possesses a two-dimensional continuum of possible steady states. As the animal traverses its environment, this continuum of steady states can be mapped in a one-to-one manner to the animal's position.

The same place cells, however, participate in representations of multiple distinct environments, collectively exhibiting global remapping [27]: the relation between activity patterns of a pair of place cells in one environment cannot predict the relation of their activity patterns in a different environment and thus each environment has its unique neural population code. To account for this phenomenon in the attractor framework, multiple, distinct continuous attractors which rely on the same place cells are embedded in the connectivity, each representing a distinct single environment [36, 5, 25, 1]. Conceptually, this is similar to the Hopfield model [18], but instead of embedding discrete memory patterns, each embedded memory pattern is a continuous manifold that corresponds to a different spatial map. Despite the network's ability to represent multiple environments, it has been argued that the embedding of multiple continuous attractors inevitably produces frozen (or quenched) noise that eliminates the continuity of steady states in each of the discrete attractors [36, 32, 19, 25, 26], leading to detrimental drifts of the neural activity. Unlike stochastic dynamical noise which is expressed in instantaneous neural firing rates, time-independent quenched noise is hard-wired in the connectivity — it breaks the symmetry between the continuum of neural representations.

From a qualitative perspective, the bump of activity can be conceptualized as residing on a minimum of an abstract energy landscape in the $N$ dimensional space of neural activity (where $N$ is the number of neurons). This energy landscape is precisely flat in the single map case, independently of the bump's position (Fig. 1a). However, contributions to the synaptic connectivity included to support additional maps distort this flat energy landscape into a similar, yet wrinkled energy landscape, and the continuous attractors are replaced by localized discrete attractor states (Fig. 1b). Consequently,

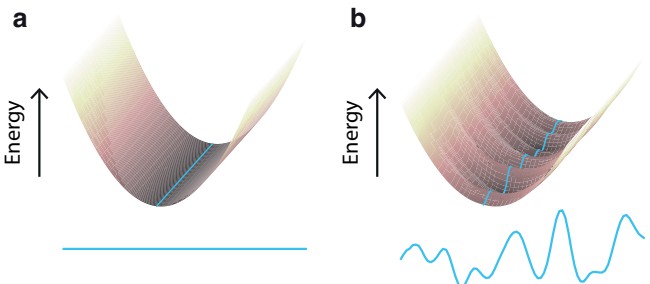

Figure 1: **Energy landscape.** Schematic illustration of energy surfaces along a 1-dimensional attractor when a single (**a**), and multiple (**b**), maps are embedded. Cyan traces show energy landscapes along bottom of surfaces.

the system can no longer stably represent a continuous manifold of bump states for any of its embedded attractors. Instead, activity patterns will systematically drift into one of the energy landscape's discrete minima, thereby degrading the network's ability to sustain persistent memory. This highly detrimental effect has been explored in previous works and was recognized as an inherent property of such networks [25, 26, 1]. External sensory inputs can pin and stabilize the representation [26, 1], but it remained unknown whether internal mechanisms could potentially attenuate the quenched noise in the system and stabilize the representations, independently of external sensory inputs. These insights raise the question, whether it is possible to embed multiple continuous attractors in a single RNN while eliminating interference effects and retaining a precisely flat energy landscape for each attractor, but without simply reducing the network capacity.

Here, we first formalize the concept of an energy landscape in the multiple attractor case. We then minimize an appropriately defined loss function to flatten the energy landscape for all embedded attractors. We show that the energy landscape can be made nearly flat, which is reflected by an attenuation of the drifts, and a dramatic improvement in the stability of attractor states across the multiple maps. These results provide a proof of principle – that internal brain mechanisms can support maintenance of persistent representations in neural populations which participate in multiple continuous attractors simultaneously, with much higher stability than previously and naively expected.

## Results

We consider the coding of spatial position in multiple environments, or *maps*, by hippocampal place cells. Within each map, the connectivity between place cells produces a continuous attractor. We adopt a simple CAN synaptic architecture with spatially periodic boundary conditions. In the one-dimensional analogue of this architecture, this connectivity maps into the ring attractor model [7, 41, 31, 49]: neurons, functionally arranged on a ring, excite nearby neighbors, while global inhibitory connections elicit mutual suppression of activity between distant neurons. This synaptic architecture leads to a bump of activity, which can be positioned anywhere along the ring.

The overall synaptic connectivity $\mathbf{J}$ is expressed as a sum over contributions from different maps, $\mathbf{J} = \sum_{l=1}^{L} J^l$, where $L$ is the number of embedded maps (see Supplementary Material, SM). To mimic the features of global remapping, a distinct spatial map is generated independently for each environment, by choosing a random permutation that assigns all place cells to a set of preferred firing locations that uniformly tile this environment. Consequently, the network possesses a discrete set of continuous ring attractors, each mapping a distinct environment to a low-dimensional manifold of population activity patterns. This is similar to previous models [36, 5, 25] but adapted here to the formalism of a dynamical rate model [1].

Since the connectivity is symmetric, it is guaranteed that the network dynamics will converge to a fixed point [10]. Qualitatively, these fixed points can be viewed as the minima of an abstract high-dimensional energy landscape in which the population activity will eventually lie at steady state: each population activity pattern is associated with an energy value, and the population dynamics are constrained to follow only trajectories that cannot increase the energy and must ultimately settle in a local minimum.

When only a single map is embedded in the connectivity, the representations span a true continuum of steady states and can be accurately read out. From the energy perspective, this continuity corresponds to a completely flat energy landscape where a continuum of steady states share an identical energy value. Initialization of the network from any arbitrary state in this case will always converge to a steady state which is a localized *idealized bump* of activity. Idealized bumps have a smooth symmetric structure, and are invariant as they can evolve at any position along the attractor.

It is sufficient, however, to embed only one additional map to distort the continuity of true steady states. Embedding multiple maps introduces quenched noise in the system which eliminates the ability to represent a true continuum of steady states in each of the attractors. From the energy perspective, the quenched noise is reflected in distortions of the energy landscape, as it becomes wrinkled with multiple minima (Fig. 1b). Thus, the discrete attractors lose their ability to represent a true continuum of steady states, and the memory patterns consequently accrue detrimental drifts (Fig. 2).

The idealized bumps from the single map case are no longer steady states of the dynamics when multiple maps are embedded. Instead, activity patterns converge on distorted versions of idealized bumps. Nevertheless, as long as the network is below its capacity, this activity is still localized. When initializing the network from an idealized bump at a random position along any of the discrete attractors, it will instantaneously become slightly distorted and will then systematically drift to the energy minimum point in its basin of attraction (Fig. 2b). These effects are enhanced as the number of embedded maps (and thus the magnitude of quenched noise) is increased and until capacity is breached when the exhibited activity patterns are not expected to be localized in any of the attractors.

### Flattening the energy landscape

To attenuate the systematic drifts that emerge upon embedding of multiple maps, we focused on flattening of the wrinkled energy landscape. The methodology throughout this work relied on the following two-step procedure: first, the wrinkled energy landscape was evaluated. Then, based on this evaluation, small modifications $\mathbf{M}$ were added to the original synaptic weights to flatten the energy landscape (SM). Importantly, the goal was to flatten the energy landscape for all maps simultaneously, which rules out the trivial solution of flattening a subset of maps by unembedding the others.

Generally, the energy value of each arbitrary state for any RNN with symmetric connectivity is given by the following Lyapunov function [10]:

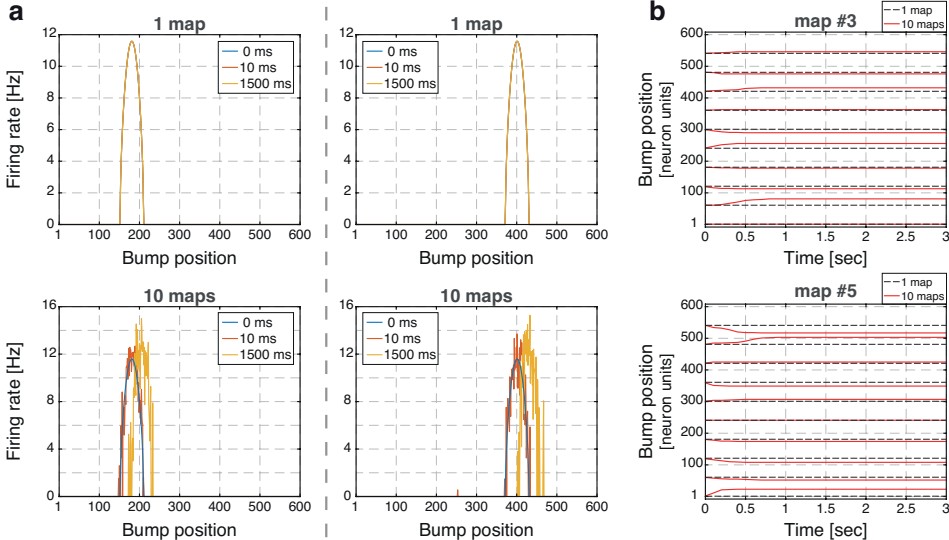

Figure 2: **Embedding multiple maps distorts the idealized bump and induces systematic drifts. a,** Two examples (left and right) showing snapshots of population activity at three time points (0 ms, 10 ms and 1500 ms). Activity was initialized at a random position along the attractor in each example. Population activity remains stationary when a single map is embedded (top), but distorts and drifts when ten maps are embedded (bottom). **b,** Superimposed bump positions versus time of ten independent idealized bump initialization (red), in two different embedded maps (top - map #3, bottom - map #5), out of ten total embedded maps. The stable bump position obtained when a single map is embedded is plotted for reference (dashed traces). Systematic drift is evident when ten maps are embedded.

$$E(\vec{I}, \mathbf{W}) = \sum_{i=1}^{N} \left( \int_{0}^{I_i} \mathrm{d}z_i z_i \phi'(z_i) - h\phi(I_i) - \frac{1}{2}\sum_{j=1}^{N} \phi(I_i)\, \mathbf{W}_{i,j}\, \phi(I_j) \right) \tag{1}$$

where $E$ is the energy value, $\vec{I}$ is the population synaptic activity, $\mathbf{W}$ is the connectivity, $h$ is an external constant current, and $\phi$ is the neural transfer function. To flatten the wrinkled energy landscape, it was first necessary to evaluate it along each of the approximated attractor manifolds. Since $h$, $\phi$, and number of neurons ($N$) are assumed to be fixed, the energy value (Eq. 1) of each state depends only on the population activity ($\vec{I}$) and the synaptic connectivity ($\mathbf{W}$) which, in our case, is decomposed into the embedded maps and the weight modifications, namely, $\mathbf{W} = \sum_l J^l + \mathbf{M}$.

We first took a perturbative approach in which we examined the idealized bumps that emerge when only a single map is embedded in the connectivity, while treating all the contributions to the connectivity that arise from the other maps as inducing a small perturbation to the energy. In Eq. (1), corrections to the energy of stationary bump states in map $l$ arise from two sources: First, a contribution arising directly from the synaptic weights associated with the other maps ($J^{l'}$ where $l' \neq l$), as well as from the modification weights $\mathbf{M}$ (third term in Eq. 1). This term, to leading order, is linear in the synaptic weights. Second, corrections arising from deformation of the bump state. Because a deformed bump drifts slowly when the perturbation to the synaptic weights is weak, it is possible to conceptually define bump states along the continuum of positions, which are nearly stationary (a precise way to do so will be introduced later on). This contribution to the energy modification is quadratic in the deformations, because the idealized bump states are minima of the unperturbed energy functional, and therefore the energy functional is locally quadratic near these minima. Hence, to leading order in the perturbation, we neglect this modification in this section. Overall, the energy $E_{\mathrm{ib}}^{k,l}$ of an idealized bump (ib) state centered around position $k$ in map $l$ is,

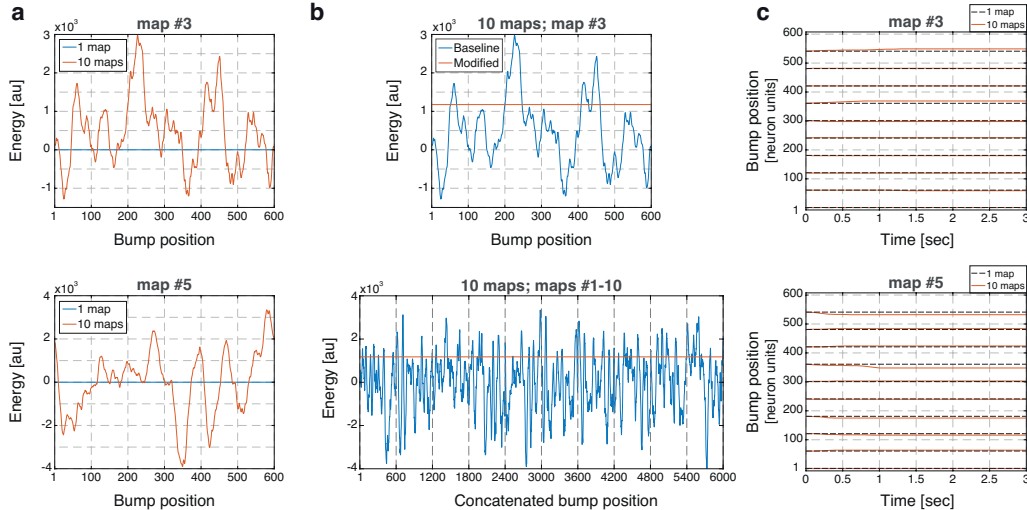

Figure 3: **Energy landscape is distorted when multiple maps are embedded but can be flattened to reduce drifts. a,** Evaluated energy landscape using idealized bumps, along two representative embedded maps (top - map #3, bottom - map #5), out of the total $L = 10$ embedded maps (orange) used in Fig. 2. The energy landscape obtained when $L = 1$ is superimposed in blue. For visibility purposes, energy landscapes are shown after subtraction of the mean across all positions and maps (SM). **b,** Re-evaluated energy landscape using idealized bumps after weight modifications were added (orange). The landscape is precisely flat. Note that this is only an approximation to the actual energy landscape, due to the use of idealized bumps (see text). Top: energy along map #3 (blue trace is identical to orange trace in panel a, top). Bottom: same as top, for all embedded maps concatenated. **c,** Bump trajectories as in Fig. 2b, but with added weight modifications. Qualitatively, the magnitude of the drifts is reduced (compare with Fig. 2b).

$$E_{\text{ib}}^{k,l} = E_0 - \frac{1}{2} \sum_{l' \neq l} \vec{r}_0^{k,l^T} J^{l'} \vec{r}_0^{k,l} - \frac{1}{2} \vec{r}_0^{k,l^T} \mathbf{M} \vec{r}_0^{k,l} \tag{2}$$

where $E_0$ is the energy value of an idealized bump state in the case of a single embedded map (which is independent of $k$ and $l$), and $\vec{r}_0^{k,l} = \phi\left(\vec{I}_0^{k,l}\right)$ is an idealized bump around position $k$ in map $l$.

As a first step, we evaluated the first two terms in Eq. 2, which represent the Lyapunov energy of idealized bump states in the absence of the weight modifications $\mathbf{M}$. In each one of the maps, the energy landscape was evaluated using these idealized bumps at $N$ uniformly distributed locations, centered around the $N$ preferred firing locations of the neurons. Since this resolution is much smaller than the width of the bump, the continuous energy landscape was densely sampled. As expected, a precisely flat energy landscape was observed when only a single map was embedded (Fig. 3a, blue traces), as the attractor is truly continuous in this case. However, when multiple maps were embedded, a wrinkled energy landscape was observed (Fig. 3a, orange traces), as a consequence of the quenched noise in the system.

We next sought small modifications in the synaptic weights which could potentially flatten the energy landscape with only mild effects on the population activity patterns. Under the approximations considered above, flattening the energy landscape requires that the right hand side of Eq. 2 is a constant, independent of $k$ and $l$. Because the energy depends on the connectivity in a linear fashion to leading order, we obtain a set of linear equations. Using the discretization described above and when $N > 2L + 1$, such a system is underdetermined with more unknown weight modifications than sampled energy evaluations along the attractors. Out of the infinite space of solutions, the least squares solution, which has the overall minimal $L_2$ norm of connectivity modifications was chosen.

As expected, re-evaluating the energy using idealized bumps with the addition of the weight modifications yielded a precisely flat energy landscape (Fig. 3b, orange traces). This does not imply, however, that drifts will vanish since we evaluated the true energy landscape only up to first order

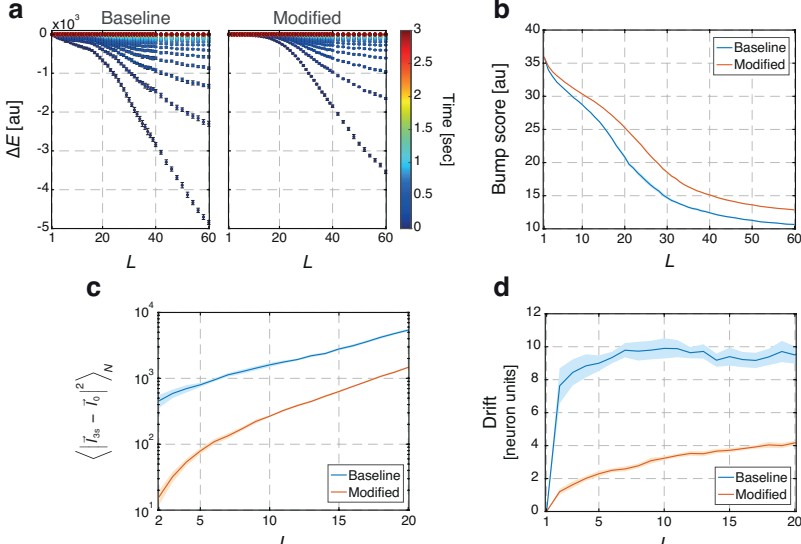

Figure 4: **Effects of weight modifications obtained using idealized bumps scheme on the stability of bump states. a,** Energy difference between consecutive network states without (left) and with (right) the weight modifications as a function of $L$, the total number of embedded maps. In both cases, the changes in energy of network states are negligible after 3 seconds, indicating that steady states were reached (see also Supplementary Fig. 1). **b,** The bump score (SM) without (blue) and with (orange) the weight modifications as a function of $L$. **c,** Mean squared change of all neuron's firing rates [Hz] between time points 0 and 3 sec, without (blue) and with (orange) the weight modifications as a function of $L$. Note that the scale is logarithmic. **d,** Measured drifts without (blue) and with (orange) the weight modifications as a function of $L$. Error bars are ±1.96 SEM in all panels (SM).

in the deviations from an idealized bump state. To test whether the weight modifications led to decreased drifts, we independently initialized the network using an idealized bump, centered in each realization at one of ten uniformly distributed positions in each attractor. First, changes in the energy of the population activity through time were monitored, to verify that steady states were reached. As expected for a Lyapunov energy function, this quantity decreased monotonically, and stabilized almost completely for all initial conditions within three seconds, indicating that the convergence to a steady state was nearly complete (Fig. 4a). To further validate that activity was nearly settled within three seconds, we measured the drift rates and the rate of mean squared change in the firing rate of all the neurons as defined below. Only subtle changes were still observed beyond this time (Supplementary Fig. 1), justifying the consideration of the states achieved after three seconds as approximated steady states. We next evaluated a *bump score* that quantifies the similarity between the population activity with any one of the idealized bumps states, placed at all possible positions along all the attractors (SM). In the large $N$ limit, a sharp transition is expected when capacity is breached, but this effect is smoothed for a smaller and finite number of neurons, as used in this study (SM). Nevertheless, adding the weight modifications did not decrease the bump score (Fig. 4b), indicating that the network can still reliably represent its multiple embedded maps. Our focus is on networks below the capacity, which we roughly estimate to be ∼20 maps for $N = 600$.

To quantify the stability of the network, we measured the mean squared change in the firing rate of all neurons, from initialization and until approximated steady states were reached (at 3s). This measure of stability improved dramatically due to the synaptic weight modifications (Fig. 4c). A second measure of stability was the drift of the bumps, obtained by examining the mean distance bumps traveled. The bump location was defined as the position that maximized the bump score. We found that the distance bumps traveled over 3s from initialization decreased significantly when the weight modifications were added to the connectivity (Fig. 4d). Since the spatial resolution of the energy landscape evaluation is identical to that in which neurons tile the environment, drifts lower than a single unit practically correspond to a perfectly stable network. These results (Fig. 4c-d) demonstrate

that flattening the approximated energy landscape across all maps by adding the synaptic weight modifications indeed increased the stability of the multiple embedded attractors.

Despite the approximation of the energy function to first order in the weight modifications, our approach achieved dramatic improvement in the network stability. Two limitations of this approach can be clearly identified: first, the energy landscape was evaluated for idealized bump states, even though these activity patterns are not the true steady states when multiple maps are embedded in the connectivity. Second, weight modifications were designed to correct the energy only up to the first order. Once introduced, these weight modifications produce slight changes in the structure of the steady states, which in turn generate higher-order corrections to the energy that were neglected in the approximation. Next, we describe how these limitations can be overcome by defining a more precise loss function for the roughness of the energy landscape.

**Iterative constrained gradient descent optimization**

As discussed above, when $L > 1$ the unmodified system does not possess true steady states at all positions: following initialization, an idealized bump state is immediately distorted, and then systematically drifts to a local minimum of the energy functional (Fig. 2). In order to define an energy landscape over a continuum of positions, it is thus necessary to first formalize the concept of states that have reached an energy minimum but also span a continuum of positions. To do so, we considered the minima of the energy functional under a constraint on the center of mass of the bump. In practice, such minima are found by idealized bump initialization, followed by gradient descent on the energy functional under the constraint. A distorted bump then dynamically evolves to minimize the Lyapunov function, while maintaining a fixed center of mass at its initial position (Supplementary Fig. 2). See SM for details on the constrained optimization procedure and its validation.

Since the goal of the constraint is to parametrize all states along the approximated continuous attractor, its exact definition is not crucial as long as the optimization is applied along a dense representation of positions. By finding the minima of the energy functional along such a dense sample of positions, it is possible to precisely measure the energy landscape of the system along each one of the attractors. As expected, the evaluated energy landscape obtained using gradient optimization achieved lower energy values compared to the energy landscape obtained using idealized bumps (Fig. 5a, orange traces).

Using this formal definition of the position dependent energy, it is possible to formulate a learning objective for the modification weights $\mathbf{M}$, designed to flatten the energy landscape. We denote by $E^{k,l}$ the constrained minimum of the energy at position $k$ in map $l$, evaluated using the constrained gradient minimization scheme described above. We approached our goal by attempting to equate the values of $E^{k,l}$ for all $k$ and $l$, using an iterative gradient-based scheme (SM). It is straightforward to evaluate the gradient of $E^{k,l}$ with respect to $\mathbf{M}$, since up to first order in $\Delta\mathbf{M}$,

$$E^{k,l}(\mathbf{M} + \Delta\mathbf{M}) - E^{k,l}(\mathbf{M}) \simeq -\frac{1}{2}\vec{R}^{k,l^T}(\Delta\mathbf{M})\,\vec{R}^{k,l} \tag{3}$$

where $\vec{R}^{k,l}$ is the firing rate population vector at the constrained minimum corresponding to $E^{k,l}$. We note that the constrained minima themselves are modified due to the change in $\mathbf{M}$, but these modifications contribute only terms of order $(\Delta\mathbf{M})^2$ to the change in the energy (see SM).

In each iteration of the weight modification scheme, $E^{k,l}$ and $\vec{R}^{k,l}$ were first evaluated numerically for all $k$ and $l$ using constrained gradient optimization and the values of $\mathbf{M}$ from the previous iteration (starting from vanishing modifications in the first iteration). Next, in iteration $i$, we sought adjustments $\Delta\mathbf{M}_i$ to the weight modifications $\mathbf{M}$, that equate the energy across all $k$ and $l$, to leading order. As in the previous section, Eq. 3 yields an underdetermined set of linear equations (assuming that $N > 2L + 1$), where the least square solution was chosen in each iteration.

Fig. 5b-c shows results obtained over several gradient optimization iterations. The mean absolute value and standard deviation of weight modifications decreased with the iteration number (Fig. 5b), and the flatness of the energy landscape improved systematically in successive iterations (Fig. 5c, top, and Supplementary Fig. 3). The standard deviation of the energy across locations and maps decreased by almost four orders of magnitude after four iterations, and the maximal deviation of the energy from its mean decreased dramatically as well (Fig. 5c, bottom).

We next examined the consequences of improved flatness of the energy landscape on the network's stability. Qualitatively, individual states exhibited much less drift after five iterations (Fig. 6a),

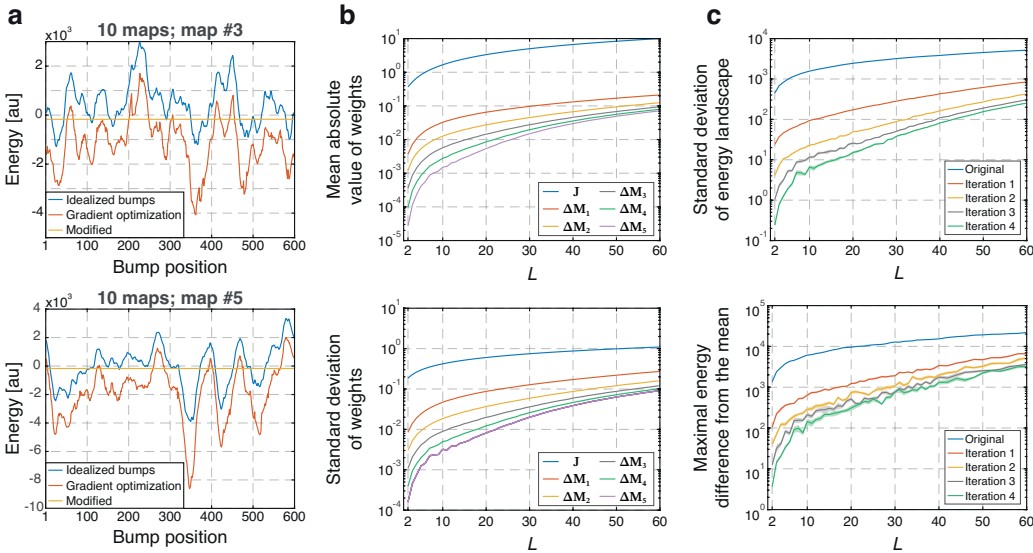

Figure 5: **Energy landscape evaluation and modifications using constrained gradient optimization. a,** Energy landscape evaluated using constrained gradient optimization (orange) for maps #3 (top) and #5 (bottom), out of $L = 10$ embedded maps as used in Figs. 2 and 3. For reference, energy landscapes evaluated using idealized bumps are plotted as well (blue traces, identical to the orange traces from Fig. 3a). As expected, the energy after gradient optimization (orange trace) is lower. Re-evaluating the energy landscape to leading order in the weight modifications yielded a precisely flat energy landscape when evaluated using the pre-modified network steady states (yellow). **b,** Top: mean absolute value of the weights in the original connectivity matrix (blue) and in modification adjustments for each gradient optimization iteration, as a function of $L$, the total number of embedded maps. Bottom: corresponding standard deviation. Error bars are ±1.96 SEM (SM). **c,** Top: standard deviation of the energy landscape without (blue) and with the weight modification for each gradient optimization iteration as a function of $L$. Bottom: same as top, but showing the peak absolute difference between all energy values and their mean. Error bars are ±1.96 SEM.

compared to the unmodified scenario (Fig. 2b), and also in comparison with the scheme based on idealized bumps (Fig. 3c). To systematically quantify the improved stability, we first validated that approximated steady states were reached three seconds after initialization (Fig. 6b), and that the bump score was not significantly affected by these modifications (Fig. 6c) as in the idealized bump approach (Fig. 4a-b). Next, we examined two measures of stability (as in Fig. 4c-d). The mean squared change in the firing rate of all the neurons, across 3s from initialization was dramatically improved, by almost three orders of magnitude, compared to the pre-modified networks for the smaller values of $L$ (Fig. 6d). Measured drifts were attenuated significantly as well (Fig. 6e). Note that, as described above, drifts which are below or equal to the discretization precision of a single unit correspond to perfect stability. For completeness, drifts were also measured using the phase of the population activity vector and demonstrated similar results (Supplementary Fig. 4). Taken together, these results show that with few constrained gradient optimization iterations, a dramatic increase can be achieved in the inherent stability of a network forming a discrete set of continuous attractors.

## Discussion

Interference between distinct manifolds is highly detrimental for the function of RNNs designed to represent multiple continuous attractor manifolds. Naively, it is sufficient to embed only a second manifold in a network to destroy the continuity of the energy landscapes in each attractor. This leads to the loss of the continuity of steady states, which are replaced by a limited number of discrete attractors. In tasks that require storage of a continuous parameter in working memory, these effects are manifested by systematic drifts that degrade the memory over time. Sensory inputs can, hypothetically, pin and stabilize the representation [26, 1], but in the absence of correcting external inputs the representation is destined to systematically drift to the nearest local energy minima, thereby

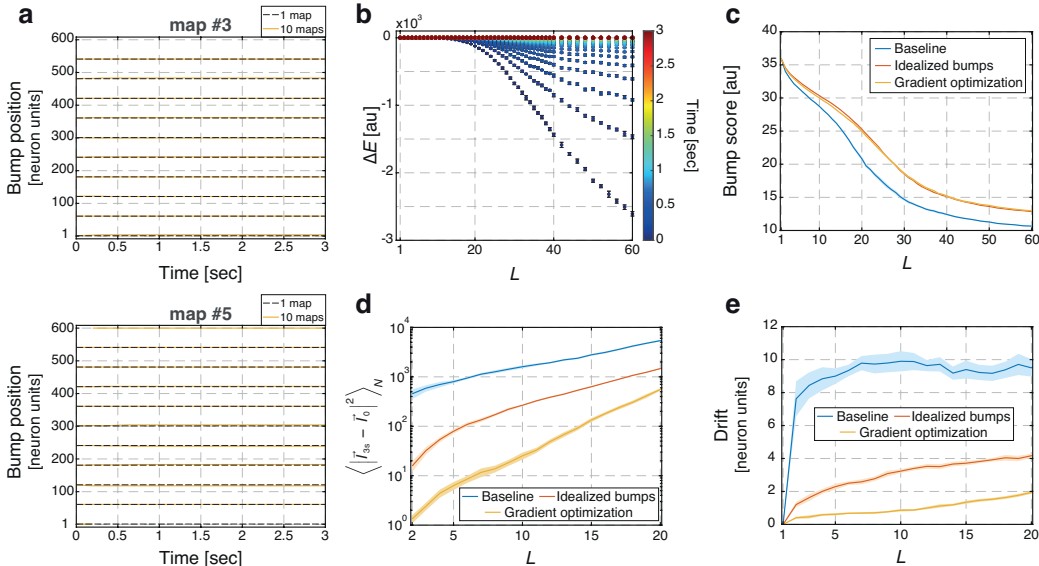

Figure 6: **Effects of weight modifications obtained using constrained gradient optimization on the stability of bump states. a,** Bump trajectories as in Figs. 2b and 3c, but with added weight modifications obtained after five constrained gradient optimization iterations. Qualitatively, the drifts almost vanish (compare with Figs. 2b and 3c, and note cyclic boundary conditions). **b,** Same as Fig. 4a, but after five iterations of the constrained gradient optimization scheme. **c-e,** Same as Fig. 4b-d, but with superimposed results obtained after five constrained gradient optimization iterations (yellow traces). Adding weight modifications did not reduce the bump score (panel c). The network stability improved significantly (panels d-e) compared to the unmodified network (blue traces) and to the modified network using the idealized bumps scheme (orange traces). Error bars are ±1.96 SEM (SM).

impairing network functionality. This has been thought to be a fundamental limitation of such networks.

Here we reexamined this question by adding small weight modifications, tailored to flatten the high dimensional energy landscape along the approximate attractor, to the naive form of the connectivity matrix. We found that appropriately chosen weight modifications can dramatically improve the stability of states along the embedded attractors. Furthermore, the objective of obtaining a flat energy landscape appears to be largely decoupled from the question of capacity, which is also limited by interference between multiple attractors. Indeed, introducing the weight modifications did not qualitatively affect the dependence of the bump score on $L$ (Fig. 6c). In particular, the kink in this function occurs at approximately $L = 20$ (for $N = 600$) both in the pre- and post-modified networks. Theoretically, the position of the kink is expected to roughly match the location of a sharp transition in the large $N$ limit, while keeping $L/N$ fixed. A recent work [6] suggests that the capacity may depend logarithmically, and thus weakly, on the prescribed density of steady states. It will be interesting to explore the relation of this result to our framework as it was obtained for RNNs composed of binary neurons, with a linear kernel support vector machine based learning rule.

We explored our schemes in 1D to reduce the computational cost, but it is straightforward to extend our approach to 2D. One notable difference between 1D and 2D environments is that the number of neurons required to achieve a good approximation to a continuous attractor, even for a single map, scales in proportion to the area in 2D, as opposed to length in 1D. However, for a given number of neurons, there is no substantial difference between the two cases in terms of the complexity of the problem: the number of equations scales as $NL$, and the number of parameters (synaptic weights) scales as $N^2$. Since the random permutations are completely unrelated to the spatial organization of the firing fields, quenched (frozen) noise is expected to behave similarly in the two cases.

For simplicity, we assumed that each cell is active in each environment, but it is straightforward to adapt the architecture to one in which the participation ratio, $p$, defined as the average fraction of maps in which each cell participates, is smaller than unity. Measurements in CA3, performed

in the same cells in multiple environments, indicate that CA3 cells are active only in a subset of environments, with $p \sim 15\%$ as a rough estimate [3]. Clearly, the quenched noise in the system increases in proportion to the number of embedded maps $L$, and it will be interesting in future work to assess how the roughness of the energy landscape depends on $N$, $L$, and $p$. We note that even though small $p$ implies low interference, it also implies a reduction in the number of neurons participating in each map, and in each bump state. This is expected to reduce the resilience of each attractor to the quenched noise. Hence, the overall effect of $p$ on the roughness of the energy landscape (when keeping $N$ fixed) is non-trivial.

Previous works have suggested that synaptic scaling [32] or short term facilitation [19, 38] mechanisms can stabilize working memory in networks with heterogeneous connectivity. These mechanisms were achieved, however, in networks which are equivalent to the single map case but with random added heterogeneity. In this respect, the source of heterogeneity in these networks is different from the one in this work, where heterogeneity arises from the embedding of multiple coexisting attractors. It would be interesting to investigate whether a synaptic facilitation mechanism [19] can be implemented in the multiple map case and further improve the network's stability alongside the energy-based approach proposed here. However, since in a discrete set of continuous attractors each neuron is participating in multiple representations, a synaptic scaling mechanism [32] is unlikely to achieve such stabilization. In addition, it may also be interesting to explore other potential methods for the design of a synaptic connectivity that supports multiple, highly continuous attractors, other than the one explored here. These might include generalization of approaches that were based on the pseudo-inverse learning rule for embedding of correlated memories in binary networks [29], or training of rate networks to possess a near-continuum of persistent states by requiring stability of a low-dimensional readout variable [12].

Our approach is based on the minimization of a loss function that quantifies the energy landscape's flatness. This raises several important questions from a biological standpoint. First, the connectivity structure obtained after training is fine-tuned, raising the question of whether neural networks in the brain can achieve a similar degree of fine tuning. A similar question applies very broadly to CAN models, yet, there is highly compelling evidence for the existence of CAN networks in the brain [45, 2, 48, 46, 39, 21, 15]. We also note that out of an infinite space of potential weight modifications only a specific set (least-squares) was chosen, leaving many unexplored solutions to the energy flattening problem which may relax the fine-tuning requirement. Second, our gradient-based learning rule for the minimization of the loss function was not biologically plausible. In recent years, however, many important insights were obtained on computation in biological neural networks by training RNN models using gradient based learning [43, 4, 33, 44, 24, 30, 20, 37, 16, 22, 42]. The (often implicit) assumption is that biological plasticity in the brain can reach similar connectivity structures as those obtained from gradient based learning, even if the learning rules are not yet fully understood. Thus, our results should be viewed as a proof of principle – that multiple embedded manifolds, previously assumed to be inevitably unstable, can be restructured through learning to achieve highly stable, nearly continuous attractors. It will be of great interest to seek biologically plausible learning rules that could shape the neural connectivity into similar structures. Such rules might be derived from the goal of stabilizing the neural representation during memory maintenance, either based solely on the neural dynamics, or on corrective signals arising from a drift of sensory inputs relative to the internally represented memory [23].

**Acknowledgments**
The study was supported by the European Research Council Synergy Grant no. 951319 ("KILONEU-RONS"), and by grant nos.1978/13, and 1745/18 from the Israel Science Foundation. We further acknowledge support from the Gatsby Charitable Foundation. Y.B. is the incumbent of the William N. Skirball Chair in Neurophysics. This work is dedicated to the memory of Mrs. Lily Safra, a great supporter of brain research.

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
