# Supplementary Material:
# Simultaneous embedding of multiple attractor manifolds in a recurrent neural network using constrained gradient optimization

**Haggai Agmon**
The Hebrew University of Jerusalem, Israel
and Stanford University, USA
`haggai.agmon@mail.huji.ac.il`

**Yoram Burak**
The Hebrew University of Jerusalem, Israel
`yoram.burak@elsc.huji.ac.il`

## Supplementary Information

### Network connectivity

The network consists of $N = 600$ neurons, which represent positions in $L$ one-dimensional periodic environments. A distinct spatial map is generated for each environment, by choosing a random permutation that assigns all place cells to a set of preferred firing locations that uniformly tile this environment.

The synaptic connectivity between place cells is expressed as a sum over contributions from all spatial maps:

$$\mathbf{J} = \sum_{l=1}^{L} J^l \tag{S1}$$

where $J_{i,j}^l$ depends on the periodic distance between the preferred firing locations of cells $i$ and $j$ in environment $l$, as follows

$$J_{i,j}^l = \begin{cases} A\exp\left[-\dfrac{\left(d_{i,j}^l\right)^2}{2\sigma^2}\right] + b & i \neq j \\ 0 & i = j \end{cases} \tag{S2}$$

The first term is an excitatory contribution to the synaptic connectivity that decays with the periodic distance $d_{i,j}^l$, with a Gaussian profile $\left(0 \leq d_{i,j}^l < N/2\right)$. The second term is a uniform inhibitory contribution. The parameters $A > 0$, $\sigma$, and $b < 0$ are listed in Network parameters. Note that the connectivity matrices corresponding to any two maps $l$ and $k$ are related to each other by a random permutation:

$$J_{i,j}^l = J_{\pi^{l,k}(i),\pi^{l,k}(j)}^k \tag{S3}$$

where $\pi^{l,k}$ denotes the random permutation from map $k$ to map $l$. Even though $\mathbf{J}$ is a $N \times N$ matrix, it has only $\left(N^2 - N\right)/2$ unique terms since $\mathbf{J} = \mathbf{J}^T$ and since $\mathbf{J}_{i,i} = 0$.

37th Conference on Neural Information Processing Systems (NeurIPS 2023).

**Dynamics**

The dynamics of neural activity are described by a standard rate model. The total synaptic current $I_i$ into place cell $i$ evolves in time according to the following equation:

$$\tau \dot{I}_i = -I_i + h + \sum_{j=1}^{N} \mathbf{J}_{i,j} \cdot \phi\left(I_j\right) \tag{S4}$$

The synaptic time constant $\tau$ is taken for simplicity to be identical for all synapses (Network parameters). The external current $h$ is constant in time and identical for all cells. This current includes two terms: $h = h_0 - (L-1)N\bar{J}\bar{R}$. The first term, $h_0$, is the baseline current required to drive activity when a single spatial map is embedded in the connectivity. In the second term, $\bar{J}$ is the average of the elements in a row of the single-map connectivity matrix, and $\bar{R}$ is the average firing rate of place cells in the bump steady state when $L = 1$. The second term compensates on average for the inputs arising from the connectivity associated with the $L-1$ maps other than the active map. It thus guarantees that the mean input to all neurons in an idealized bump state will be independent of the number of embedded maps.

The transfer function $\phi$ determines the firing rate (in Hz) of place cells ($r$) as a function of their total synaptic inputs. To resemble realistic neuronal F-I curves, it is chosen to be sub-linear:

$$\phi\left(x\right) = \begin{cases} 0 & x \leq 0 \\ \sqrt{x} & x \geq 0 \end{cases} \tag{S5}$$

Note that $\phi'\left(x\right) \geq 0 \ \forall x$, which implies that the Lyapunov energy (Eq. S8) cannot increase. We implemented the dynamics using the Euler-method for numeric integration, with a time step $\Delta t$ (Network parameters).

**Network parameters**

| | |
|---|---|
| $A$ | 0.665 Hz |
| $\sigma$ | 15 neuron units |
| $b$ | $-0.2076$ Hz |
| $\tau$ | 15 ms |
| $\Delta t$ | 0.2 ms |
| $h_0$ | 10 Hz$^2$ |

**Bump score and location analysis**

To identify whether the place cell network expresses a bump state, and to identify its location $x$ and associated spatial map $l$, we define an overlap coefficient $q^l\left(x\right)$ that quantifies the normalized overlap between the population activity pattern and the activity pattern corresponding to position $x$ in spatial map $l$:

$$q^l\left(x\right) = \sum_i P_i^l\left(x\right) \cdot \hat{r}_i \tag{S6}$$

where $\hat{r}_i$ is the normalized firing rate of place cell $i$ such that the maximal firing rate across all cells is 1, and $P_i^l\left(x\right)$ is the normalized firing rate of neuron $i$ in an idealized bump state localized at position $x$ in map $l$. The idealized bump (as defined above) is obtained from the activity of a network in which a single map (map $l$) is embedded in the neural connectivity, and therefore there is no quenched noise. This definition of the bump score is similar to use of the overlap measure between the memory pattern and the network state in the theory of Hopfield networks.

Next, we define a bump score for each spatial map, defined as the maximum of $q^l\left(x\right)$ over all positions $x$ in spatial map $l$:

$$Q^l = \max_x q^l\left(x\right) \tag{S7}$$

Finally, the map with the highest $Q^l$ value is considered as the winning map, and the location $x$ that generated that value is considered as the location of the place cell bump within that map.

**Connectivity modifications to flatten the energy landscape**

The Lyapunov energy depends on the total synaptic current of all neurons, represented by the vector $\vec{I}$, and on the synaptic connectivity $\mathbf{J}$ as follows (Cohen and Grossberg, 1983),

$$E(\vec{I}, \mathbf{J}) = \sum_{i=1}^{N} \left( \int_0^{I_i} \mathrm{d}z_i z_i \phi'(z_i) - h\phi(I_i) - \frac{1}{2} \sum_{j=1}^{N} \phi(I_i) \mathbf{J}_{i,j} \phi(I_j) \right) \tag{S8}$$

The evaluation of the energy landscape was performed for all attractors at uniformly distributed positions where the synaptic activities $\vec{I}$ were centered around place cell preferred firing positions. These synaptic activities were either idealized bumps or the converged activities obtained through constrained gradient optimization (described in the next subsection).

After evaluating the energy landscape, we sought to find a connectivity modification matrix $\mathbf{M}$, such that its addition to the connectivity $\mathbf{J}$ will result in an identical energy value for each activity pattern $\vec{I}^{k,l}$ used for the energy landscape evaluation ($\vec{I}^{k,l}$ represents the population activity which encodes the $k$'th place cell preferred firing position in the $l$ map). Thus, we demanded that

$$E\left(\vec{I}^{k,l}, \mathbf{J} + \mathbf{M}\right) = C \tag{S9}$$

where $C$, is an unknown constant energy value. Note that only the third term of Eq. S8 directly depends on the connectivity $\mathbf{J}$, and since this dependence is linear, Eq. S8 can be decomposed into

$$E(\vec{I}, \mathbf{J}+\mathbf{M}) = \sum_{i=1}^{N} \left( \underbrace{\int_0^{I_i} \mathrm{d}z_i z_i \phi'(z_i)}_{T_1} - \underbrace{h\phi(I_i)}_{T_2} - \underbrace{\frac{1}{2} \sum_{j=1}^{N} \phi(I_i) \mathbf{J}_{i,j} \phi(I_j)}_{T_3} - \underbrace{\frac{1}{2} \sum_{j=1}^{N} \phi(I_i) \mathbf{M}_{i,j} \phi(I_j)}_{T_4} \right) \tag{S10}$$

To comply with the properties of $\mathbf{J}$, we demanded that $\mathbf{M} = \mathbf{M}^T$, and $\mathbf{M}_{i,i} = 0$. Hence, the number of unknown weight modifications was $\frac{N^2-N}{2}$. Consequently, writing the modification term $T_4$ at the evaluated $k$ place cell position in map $l$ (i.e., for activity $\vec{I}^{k,l}$) yields,

$$T_4^{k,l} = -\frac{1}{2} \vec{\phi}^{k,l} \cdot \mathbf{M} \cdot \vec{\phi}^{k,l^T} = -\sum_{i=1}^{N} \sum_{j>i} \mathbf{M}_{i,j} \phi_i^{k,l} \phi_j^{k,l} \tag{S11}$$

where $\phi^{k,l}$ is the transfer function output vector of the synaptic activity vector $\vec{I}_{k,l}$ which encodes the $k$'th place cell preferred firing position in the $l$ map. By rearranging only the desired $\frac{N^2-N}{2}$ elements of $\mathbf{M}$ into a column vector $\vec{m}$, and noting that Eq. S11 defines a set of $N \cdot L$ linear equations for these elements, we can rewrite Eq. S11 as

$$\mathbf{A}\vec{m} = \vec{\kappa} \tag{S12}$$

where $\mathbf{A}$ is a matrix with $N \cdot L$ rows and $\frac{N^2-N}{2}$ columns, and $\vec{\kappa}$ is the corresponding vector of solutions ($\vec{\kappa} = C - [T_1 + T_2 + T_3]$ evaluated in each element of $\vec{\kappa}$ for a specific combination of $l$ and $k$).

Subtracting the first equation from all equations yields a similar system of $N \cdot L - 1$ linear equations but which is independent of $C$. This system is underdetermined ($\forall\ N \geq 2L+1$), and the least square solution for the sought weight modifications is given by

$$\vec{m} = \mathbf{A}^T \left( \mathbf{A}\mathbf{A}^T \right)^{-1} \vec{\kappa} \tag{S13}$$

Rearranging $\vec{m}$ back into the corresponding elements in the matrix $\mathbf{M}$ and updating the connectivity, $\mathbf{J} + \mathbf{M} \to \mathbf{J}$, must yield a precisely flat energy landscape when evaluated using the same activities $\vec{I}^{k,l}$ that were previously used to evaluate the energy landscape without the weight modifications. Moreover, since the original state was a minimum of the Lyapunov energy function with respect to the activities, deviations in the activities contribute only quadratically to the energy. Consequently, the procedure described above equalizes the energy function across all states, to linear order in the weight modifications.

**Iterative constrained gradient optimization**

In the iterative optimization scheme, the unstable idealized bumps were replaced with the converged states obtained through a constrained optimization. These states exhibit distorted activity patterns and lie at a local energy minimum subject to a constraint on the center of mass (position) of the bump. These states are found by descending through the energy landscape, while constraining the population activity bump to a fixed position.

To implement the constrained optimization, the objective function of the synaptic activities $H\left(\vec{I}\right)$ is defined as follows,

$$H\left(\vec{I}\right) = E\left(\vec{I}, \mathbf{J}\right) + \lambda \left[ f\left(\vec{I}\right) - f_0 \right] \tag{S14}$$

where $\lambda$ is an adjusting Lagrange multiplier, the function $f$ (specified next) returns the position (center of mass) associated with $\vec{I}$, and $f_0$ is the encoded position of activity at initialization at timestep $t = 0$. We seek all converged states $\vec{I}^k$ which minimize $H$ while encoding all possible place cell preferred firing positions. In the first iteration, these states are achieved by initializing the network from idealized bump (ib) states $\vec{I}_{\text{ib}}^k$ centered around all possible place cell preferred firing positions in all the embedded maps. In the following iterations, the network is initialized with the corresponding converged states from the preceding iteration. From each such initial state, small updates in the activity $\overrightarrow{\Delta I}$ are added to the activity which minimize the energy value but without changing the output of $f$. The time evolution of activity states is written as

$$\vec{I}_{t+1} = \vec{I}_t + \overrightarrow{\Delta I}_t \tag{S15}$$

The small changes in the activity $\overrightarrow{\Delta I}_t$ are determined through a gradient descent scheme where the objective function is differentiated with respect to $\vec{I}$,

$$\overrightarrow{\Delta I}_t = -\eta \cdot \overrightarrow{\nabla} H = -\eta \sum_{i=1}^N \frac{\partial E}{\partial I_i} \hat{e}_i - \eta \lambda \sum_{i=1}^N \frac{\partial f}{\partial I_i} \hat{e}_i \tag{S16}$$

where $\eta > 0$ is the learning rate and the $\hat{e}_i$'s are standard unit vectors spanning an orthonormal basis. The change in the output position of $f$ due to changes in the activity is written as

$$\Delta f_t = \sum_{j=1}^N \frac{\partial f}{\partial I_j} \hat{e}_j \cdot \overrightarrow{\Delta I}_t \tag{S17}$$

However, since the constraint enforces that there is no change in the encoded position during such gradient steps, we demand that $\Delta f_t = 0$. Plugging the expression for $\overrightarrow{\Delta I}_t$ from Eq. S16 yields

$$\sum_{j=1}^N \frac{\partial f}{\partial I_j} \hat{e}_j \cdot \left[ \eta \sum_{i=1}^N \frac{\partial E}{\partial I_i} \hat{e}_i + \eta \lambda \sum_{i=1}^N \frac{\partial f}{\partial I_i} \hat{e}_i \right] = 0 \tag{S18}$$

For convenience, we omitted the timestep $t$ index notation as it is shared with all terms from here onward. The Lagrange multiplier, calculated at each timestep, is thus

$$\lambda = -\frac{\sum_{i=1}^{N} \frac{\partial E}{\partial I_i} \cdot \frac{\partial f}{\partial I_i}}{\sum_{i=1}^{N} \left(\frac{\partial f}{\partial I_i}\right)^2} \tag{S19}$$

To obtain an explicit expression for $\lambda$, we conclude by evaluating $\frac{\partial E}{\partial I_i}$ and $\frac{\partial f}{\partial I_i}$.

Since the connectivity is symmetric, the derivative of the energy with respect to $I_i$ is given by

$$\frac{\partial E}{\partial I_i} = I_i \phi'(I_i) - h\phi'(I_i) - \frac{1}{2}\sum_{a=1}^{N} \phi'(I_i)\mathbf{J}_{i,a}\phi(I_a) - \frac{1}{2}\sum_{a=1}^{N} \phi(I_a)\mathbf{J}_{a,i}\phi'(I_i)$$

$$= \phi'(I_i)\left[I_i - h - \sum_{a=1}^{N} \mathbf{J}_{i,a}\phi(I_a)\right] \tag{S20}$$

To differentiate the constraint with respect to $I_i$, we first define the function $f\left(\vec{I}\right)$. For simplicity, this function is defined in terms of the weighted averaged position of the rates $\vec{r} = \phi\left(\vec{I}\right)$, where the weights represent the position of the neurons relative to a reference position $x_r$ in the vicinity of the bump (the need to measure displacements relative to a reference position arises due to the periodic boundary conditions). For simplicity, the reference position is chosen as the one that maximizes the bump score. The displacement of the bump from this reference position is evaluated as

$$S\left(\vec{r}\right) = \frac{\sum_{i=1}^{N} \Delta x_i \cdot r_i}{\sum_{i=1}^{N} r_i} \tag{S21}$$

where $\Delta x_i$ is the displacement of neuron $i$ from the reference position $x_r$, defined using periodic boundary conditions such that it lies in the range $[-N/2, N/2]$. Finally,

$$f\left(\vec{I}\right) = x_r + S \tag{S22}$$

Differentiating the constraint yields

$$\frac{\partial f}{\partial I_i} = \frac{\partial S}{\partial r_i} \cdot \frac{\partial r_i}{\partial I_i} = \phi'(I_i) \cdot \frac{x_i \sum_k r_k - \sum_j r_j x_j}{\left(\sum_k r_k\right)^2} \tag{S23}$$

Plugging back Eqs. S20 and S23 in Eq. S19 will yield the required Lagrange multiplier for each time step.

**Gradient of constrained energy with respect to weights**

To derive Eq. 3 we evaluate the derivative of $E^{k,l}$ with respect to the elements of $\mathbf{M}$. Recall that $I^{k,l}$ is the state that minimizes the energy (Eq. 1) under a constraint on the position of the bump, and that $E^{k,l}$ is the energy associated with this state. Thus, both $I^{k,l}$ and $E^{k,l}$ depend on the weights $\mathbf{M}$. The gradient of $E^{k,l}$ with respect to the weights can be obtained from Eq. 1, while taking into account both the direct dependence of the energy on $\mathbf{M}$, and the implicit dependence arising from the influence of $\mathbf{M}$ on the state $\vec{I}^{k,l}$:

$$\frac{\partial E^{k,l}}{\partial \mathbf{M}_{ij}} = -\frac{1}{2}\phi\left(I_i^{k,l}\right)\phi\left(I_j^{k,l}\right) + \nabla E^{k,l} \cdot \frac{\partial \vec{I}^{k,l}}{\partial \mathbf{M}_{ij}} \tag{S24}$$

where $\nabla$ represents a gradient with respect to $\vec{I}^{k,l}$, and $\vec{R}^{k,l} = \phi\left(\vec{I}^{k,l}\right)$. The first term on the right hand side is the contribution to the derivative with respect to $\mathbf{M}_{ij}$ arising from the explicit dependence of the Lyapunov function on $\mathbf{M}$. This term is equivalent to the second term in Eq. 3. The second term on the right hand side of Eq. S24 represents the contribution to the change in the Lyapunov energy, arising from the change in the steady state activity pattern $\vec{I}^{k,l}$ in response to an infinitesimal modification of of $\mathbf{M}_{ij}$. Since $\vec{I}^{k,l}$ obeys a constraint on the center of mass of the bump for any choice of the weights, its derivative with respect to $\mathbf{M}_{ij}$ is in a direction in the $N$ dimensional neural activity space in which the center of mass is kept fixed. On the other hand, the energy has been minimized precisely within that subspace, and therefore its gradient with respect to $\vec{I}^{k,l}$, projected on this direction, must vanish (see also Supplementary Fig. 2). Hence, the second term in Eq. S24 vanishes, and up to linear order in $\Delta\mathbf{M}$ only the explicit dependence of the energy on $\mathbf{M}$ contributes to the gradient, as stated by Eq. 3.

**Energy landscape shifts**

Energy landscapes were uniformly shifted throughout the manuscript by a constant (Figs. 3a-b and 5a) in order to allow for the landscapes to be shown on the same plot for a single map and for 10 maps (Fig. 3a). This constant was selected separately for each value of embedded maps $L$ such that the mean energy of idealized bump states across all states and maps (without any weight modification) is zero: thus, the mean of the blue trace in the bottom panel of Fig. 3b is zero (as well as single map blue traces, Fig. 3a). Uniform shifts of the energy landscape are inconsequential for the stability and dynamics of the bump states and so this was consistently performed solely for visibility purposes of Fig. 3a.

**Statistical analysis**

For each network with a different number of total embedded maps, 15 realizations were performed in which the permutations between the spatial maps were chosen independently and at random. Standard errors of the mean (SEM) were evaluated across these independent realizations. Error bars in Figs. 4, 5, 6 and Supplementary Figs. 1 and 4 are ±1.96 SEM, corresponding to 95% confidence interval.

**Code availability**

Code is available at public repository `https://doi.org/10.5281/zenodo.10016179`.

# Supplementary Figures

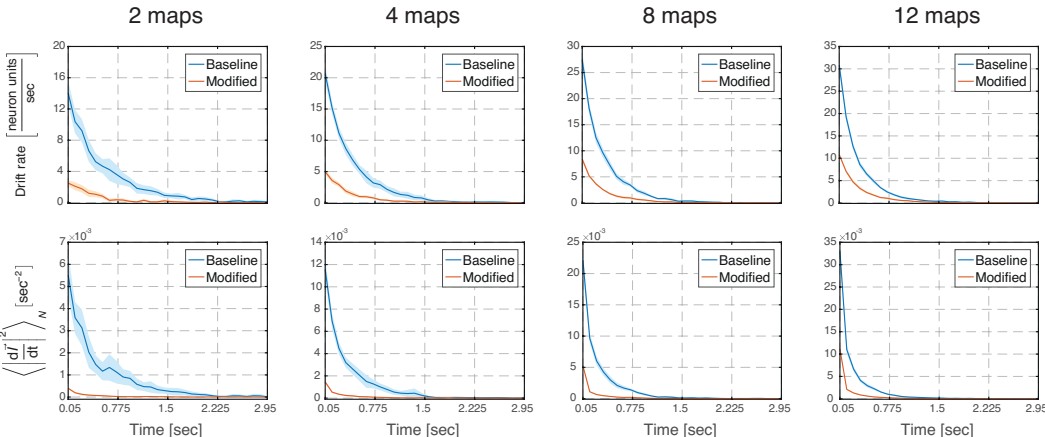

Supplementary Figure 1: **Only negligible dynamic changes persist in population activity three seconds from initialization.** Drift rates (top) and the rate of mean squared change in the firing rate of all the neurons (bottom), without (blue) and with (orange) the weight modifications as a function elapsed time since initialization for various total number of embedded maps (2, 4 ,8 and 12). Activity can be approximated as settled to steady state 3 seconds from initialization. Error bars are ±1.96 SEM.

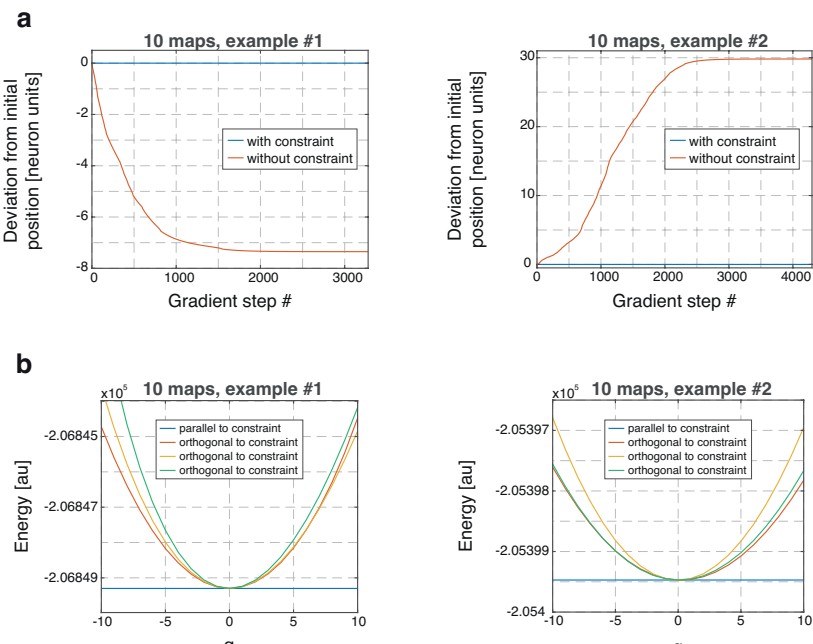

Supplementary Figure 2: **Bump position remains fixed and reaches a local energy minimum during the gradient optimization. a,** Two examples showing the relative bump position during a constraint (blue) and unconstrained (orange) gradient optimization. The bump position remains fixed during constrained optimization and systematically drifts during unconstrained optimization. **b,** The energy along corresponding projections of representative population activity vectors in the $N$ dimensional space after gradient optimization has terminated (the scalar $\alpha$ multiplies the multidimensional activity vectors). The energy along directions which are parallel to the constraint (blue) are flat while the energy along directions which are orthogonal to the constraint are quadratic (orange, yellow, and green traces. $R^2 = 0.99$).

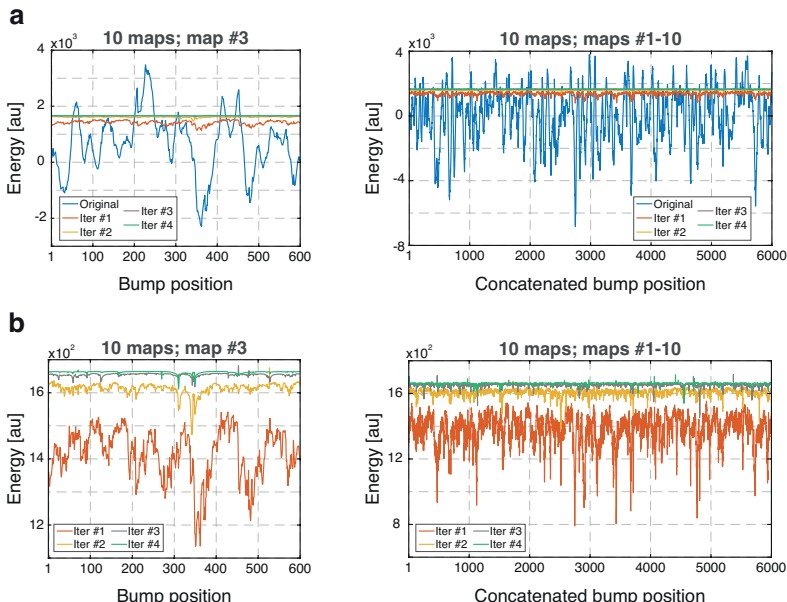

Supplementary Figure 3: **Energy landscape is flattened with increased gradient optimization iterations. a,** Evaluated energy landscapes at each optimization iteration for embedded map #3 (left), out of the total of ten embedded maps as used in Fig. 2 and for all embedded maps (right) concatenated. **b,** Same as (a) but without plotting the first gradient evaluation, to emphasize the flattening of the energy landscape as the number of optimization iterations increases.

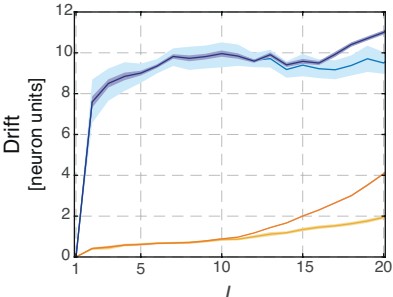

Supplementary Figure 4: **Measured drifts using weighted population activity vector phase.** Same as Fig. 6e (light blue and yellow traces) but with superimposed measured drifts using the phase of the population activity vector (dark blue and orange traces). Very similar results are obtained using the two methods, up to $L \approx 12$ embedded maps. The bump position obtained using the weighted population activity phase is noisy as all the neuron's activities contribute to the inference of the bump position using the population vector phase. Therefore, as the load increases, neurons which are in the periphery of the bump that start to fire contribute to an inaccurate inference of the bump position, leading to larger measured drifts than those observed when using the bump score. Error bars are ±1.96 SEM.