# OpenReview forum: "Simultaneous embedding of multiple attractor manifolds in a recurrent neural network using constrained gradient optimization"
_NeurIPS.cc/2023/Conference — NeurIPS 2023 poster_

### Official Review · Reviewer_MLg2 · 2023-06-30

**Soundness:** 3 good
**Presentation:** 3 good
**Contribution:** 4 excellent
**Rating:** 7
**Confidence:** 2

**Summary:**

The paper investigates the challenge of embedding multiple continuous attractor manifolds within a single RNN, with a focus on hippocampal place cells. The issues arise due to the presence of discrete steady states, visualized as minima on an abstract energy landscape, which disrupt the continuity of network activity patterns. This disruption prompts systematic drift of population activity patterns towards these discrete states, resulting in degraded memory over time. Past studies have considered the stabilizing influence of external stimuli; however, solutions in their absence remain unclear. The authors address this issue by modifying the synaptic weight to flatten the energy landscape, showcasing through simulations how this significantly stabilizes the activity pattern.

**Strengths:**

o	The methodology and problem formulation are clearly articulated.

o	Simulations indicate that weight modification significantly improves the stability of activity patterns.

o	The authors provide the code in the appendix.


**Weaknesses:**

o	While the manuscript is generally well-written, some areas could benefit from improvement. A diagrammatic illustration could help elucidate the issue of discretized states in the context of the energy landscape. Further details of the simulations (e.g., duration and specific inputs to the network) need to be provided. The appendix, which helps clarify the methodology, should be referred to more often in the Results section.

o	Strengthening the paper could involve additional simulations and/or discussions (see questions below).

o	There is a minor typo: "emfbedded map" should be "embedded map."


**Questions:**

o	How applicable is your approach if connection weights are not symmetrical?

o	How robust is the pattern stability if the modified weights are slightly perturbed?

o	Can your approach provide biological insights into the structure of weight modification when new maps are added?

o	Does the resulting weight (after the modification) violate Dale’s law?

o	While it's evident that an energy landscape embedded with minima leads to activity patterns settling into one of them, thereby degrading the network’s ability to sustain persistent memory, does a flat energy landscape guarantee pattern stability?

o	Here are some additional related works that could be included in the discussion [1-2].

o	References:

	[1] Genkin and Engel, Nature Machine Intelligence, 2020: https://www.nature.com/articles/s42256-020-00242-6

	[2] Whittington et al., Cell, 2020:
https://www.sciencedirect.com/science/article/pii/S009286742031388X


**Limitations:**

The authors have acknowledged certain limitations, such as the unclear biological plausibility of their approach.

---

> ### Author Rebuttal · Authors · 2023-08-07
>
> >While the manuscript is generally well-written, some areas could benefit from improvement. A diagrammatic illustration could help elucidate the issue of discretized states in the context of the energy landscape.
>
> We agree that it will be helpful to include a schematic illustration to clarify the concept of discrete minima in the energy landscape. Since the final version can include one additional page, we plan to include such illustration if accepted. We thank the reviewer for suggesting this.
>
> >Further details of the simulations (e.g., duration and specific inputs to the network) need to be provided.
>
> The Supplementary text includes full details of the network dynamics, parameters, the numerical integration scheme used to simulate the dynamics, the optimization scheme, and quantifications. We will be grateful if the reviewer would let us know in case something specific seems to be missing. Note that there are no training inputs in our study.
>
> >The appendix, which helps clarify the methodology, should be referred to more often in the Results section.
>
> We thank the reviewer for this comment and now refer to the Supplementary Material more frequently in the Results section.
>
> >Strengthening the paper could involve additional simulations and/or discussions (see questions below).
>
> See responses below.
>
> >There is a minor typo: "emfbedded map" should be "embedded map."
>
> Fixed, thanks!
>
> >How applicable is your approach if connection weights are not symmetrical?
>
> Our approach is based on the existence of a Lyapunov function [9], and this relies on symmetric connections. All existing models for embedding multiple attractors in CA3 adhere to this rule. Furthermore, one of the key values of our work, in our view, is that it provides a proof of principle for the existence of a more stable version for a discrete set of continuous attractors than previously expected, which was assumed to be inevitably unstable. In future work, it will be highly interesting to seek biologically plausible learning rules, and to relax the assumption of symmetric connectivity.
>
> >How robust is the pattern stability if the modified weights are slightly perturbed?
>
> One of the main criticisms of continuous attractor networks is that they require fine tuning of the weights: small perturbations to the weights destroy the continuity of the attractor. This is also why the embedding of multiple maps, using the standard prescription, leads to a wrinkled energy landscape. Considering this, it is a priori expected that the recovery of a flat energy landscape would rely on weights that require fine tuning. One of the interesting features of our results is that the required correction weights are small (see Fig. 4b). A hypothetical learning rule that is based on stability of the attractor states would thus need to explore only small local perturbations to the initial weights, whereas the initial weights could be learned based on a simple Hebbian rule. Even though the correction weights require fine tuning along some dimensions, note that out of an infinite space of potential weight modifications only a specific set (least-squares) was chosen, leaving many unexplored solutions to the energy flattening problem which may relax the fine-tuning requirement to some extent.
>
> >Can your approach provide biological insights into the structure of weight modification when new maps are added?
>
> See above response. The main correction to the weights is the one arising from the naive embedding scheme. The modifications that we identify in our study are rather subtle. Therefore, attempts to relate plasticity under exposure to new maps to theoretical constructs should first focus on the predictions of the naive scheme.
>
> >Does the resulting weight (after the modification) violate Dale’s law?
>
> Like many theoretical models of attractor dynamics in the hippocampus (and elsewhere in the brain) we abstract away the distinction between excitatory and inhibitory connections. One imagines that inhibitory connections are realized via interneurons.
>
> The unmodified synaptic connections in our model can be thought of as consisting of all-to-all inhibitory connections (mediated by interneurons), and excitatory connections which are specific to neuron pairs that have similar tuning in any one of the maps. Therefore, it is easy to reformulate the model in accordance with Dale’s law.
>
> Since in our model each neuron forms ~100 excitatory connections associated with each map, almost all neurons end up being connected to each other with an excitatory connection when the number of embedded maps exceed 6 (since N=600). The small modifications to the weights that are required to flatten the energy landscape could thus be implemented as a subtle increase (or decrease) in the strength of the excitatory weights. Alternatively, the weight modifications could be applied to both excitatory and inhibitory synapses. It will be interesting to flesh out these ideas in a follow-up work.
>
> >While it's evident that an energy landscape embedded with minima leads to activity patterns settling into one of them, thereby degrading the network’s ability to sustain persistent memory, does a flat energy landscape guarantee pattern stability?
>
> Yes. The existence of a Lyapunov function guarantees that dynamics will settle on local minima of the function. When minima form a manifold (a continuous set of states, all sharing the same energy), stability of each state along this manifold is guaranteed. The property of this Lyapunov function is that whenever dI/dt≠0 → dE/dt<0, and thus a flat energy landscape guarantees pattern stability.
>
> >Here are some additional related works that could be included in the discussion [1-2].
> References:
> [1] Genkin and Engel, Nature Machine Intelligence, 2020: https://www.nature.com/articles/s42256-020-00242-6
> [2] Whittington et al., Cell, 2020: https://www.sciencedirect.com/science/article/pii/S009286742031388X
>
> Thanks, we will incorporate these references.

---

> > ### Comment · Reviewer_MLg2 · 2023-08-13
> > **Response acknowledged**
> >
> > Thank you for your thorough response and additions to the paper. I am going to increase my score.

---

### Official Review · Reviewer_fsQk · 2023-07-05

**Soundness:** 4 excellent
**Presentation:** 4 excellent
**Contribution:** 3 good
**Rating:** 7
**Confidence:** 4

**Summary:**

The paper studies the storage of multiple continuous attractors in a recurrent neural network. Specifically, the authors tackle the interference between attractors and its effect on activity bump drift. By using a perturbative approach, they compute a correction to the connectivity that reduces the drift dramatically.
Continuous attractors (e.g., ring model) are important models in neuroscience, and understanding them is an important task. Multiple attractors are relevant, for example, in the CA3 region of the hippocampus, where remapping of place cells in different environments is common. Nevertheless, a naïve connectivity that is a superposition of several ring-connectivities results in only approximate continuous attractors. The result is a few stable points in each attractor, to which dynamics converge.
The authors use the perspective of an energy function (Lyapunov), and examine how the interference renders this function non-flat. They then calculate the perturbation to leading order, and solve for a change in connectivity that will flatten the energy. Furthermore, using gradient descent, they are able to achieve even greater precision.


**Strengths:**

Continuous attractors are a fundamental building block in the study of recurrent neural networks in neuroscience contexts. There are relatively few studies tackling multiple such attractors. The method to reduce interference is novel.

**Weaknesses:**

First, as the authors note, the resulting connectivity is extremely fine-tuned. This is a known problem with continuous attractors that is not addressed here.
Second, the problem and solution are highly related to similar problems in discrete attractors. Interference in Hopfield networks was tackled using pseudo inverse rules, either approximated online or as a global formula. There is no discussion of the relation to these works. The SVM approach of Battista and Monasson (Ref 5) is perhaps a similar example in continuous attractors.


**Questions:**

L146 energy IS quadratic
L178 “vast majority” the sentence isn’t very clear. Was the intention “close to a fixed point” or something similar?
L146 – I didn’t fully understand the argument on why the first order vanishes.
Can you say something about higher dimensions? Place cells are relevant in 2D. The results of [5,23] suggest differences when dimension increases.


**Limitations:**

yes

---

> ### Author Rebuttal · Authors · 2023-08-07
>
> >First, as the authors note, the resulting connectivity is extremely fine-tuned. This is a known problem with continuous attractors that is not addressed here.
>
> We agree (please see also our 7'th response to reviewer MLg2, regarding fine-tuning).
>
> >Second, the problem and solution are highly related to similar problems in discrete attractors. Interference in Hopfield networks was tackled using pseudo inverse rules, either approximated online or as a global formula. There is no discussion of the relation to these works. The SVM approach of Battista and Monasson (Ref 5) is perhaps a similar example in continuous attractors.
>
> The potential relation to the works on pseudo-inverse based variants of the Hopfield network architecture is interesting, and we thank the reviewer for bringing this up. While it will be interesting to consider this relation in depth, we would like to point out two potential differences from the scheme explored in our work: First, the goal of the pseudo-inverse rule is to precisely embed a set of prescribed patterns as attractor states of the dynamics, whereas we are willing to accept distortions in the bump states and only care about (i) keeping them localized and (ii) precisely equalizing their energy. We suspect that an attempt to embed the idealized bumps as perfect steady states (using perhaps a generalization of the pseudo-inverse rule) would prove more difficult than our goal, yet this is an interesting question and our intuition on this is only an initial thought. The second difference is that the key benefit of the pseudo-inverse method is in increasing the capacity of the network. There are some conceptual issues in relating a definition of capacity in our problem to the definition in the discrete and binary case, but if we judge the capacity by the number of maps at which delocalized states start to appear it seems that our approach does not strongly affect the capacity (Fig. 5c), whereas its main outcome is in the flattening of the energy landscape.
>
> We now briefly mention in the Discussion the potential interest in exploring a generalization of the pseudo-inverse rules, as a means of enhancing the stability of states along simultaneously embedded continuous attractors.
>
> >L146 energy IS quadratic
>
> Fixed, thanks.
>
> >L178 “vast majority” the sentence isn’t very clear. Was the intention “close to a fixed point” or something similar?
>
> Yes, this was the intention, and we thank the reviewer for pointing out that the phrasing was unclear.
>
> We now revised it by: “...indicating that the convergence to a steady state was nearly complete (Fig. 3a).”
>
> >L146 – I didn’t fully understand the argument on why the first order vanishes.
>
> We now revised the sentence in lines 146-148 as follows:
>
> “The precise structure of these states affects the second and third terms on the right hand side of Eq. (2) via the deviation of the state from an idealized bump. This contribution to the energy modification is quadratic in the deformations, because the idealized bump states are minima of the unperturbed energy functional, and therefore the energy functional is locally quadratic near these minima.”
>
> We hope that this helps clarify the argument. Please let us know if further clarification is required.
>
> >Can you say something about higher dimensions? Place cells are relevant in 2D. The results of [5,23] suggest differences when dimension increases.
>
> Ref. [23] has shown that unlike in 1D, activity bumps can bypass an energy barrier in 2D when a constant external input (or force) is applied. Intuitively, this happens since in 1D there is only a single possible direction for the bump’s motion along the applied force while in 2D there may be multiple directions (each with an overall smaller force), in which the bump could bypass the energy barrier. The question of bypassing energy barriers given external inputs is, however, decoupled from the question of systematic drifts that we address here, which are independent of external inputs. Comparing, perhaps, the typical magnitude of these systematic drifts in a naive 1D vs 2D network is indeed interesting and will be explored in future work. Nevertheless, the proof of principle provided here in 1D should be easily extended to 2D, and so it is expected that emerging systematic drifts in a naive 2D network will be attenuated using the methodology we demonstrated in 1D.
>
> There are N true steady states in a continuous attractor formed by N neurons, independently from a 1D or 2D organization. In order to produce an approximate continuous attractor, these states should densely tile the represented space, and this implies a different scaling with the size of the environment in 2D (quadratic) vs 1D (linear). Once the number of neurons is sufficiently large to allow for the single-map attractor to be nearly continuous, we do not see any conceptual difference in the problem of flattening of the energy landscapes for simultaneously embedded attractors between 1D and 2D. In both cases, the number of equations that need to be solved scales as N*L, whereas the number of weights scales as N^2. While the flattening of the energy landscape is expected to behave very similarly, we do not know how the number of true minima of the energy landscape (which is never precisely flat, even in our scheme) will behave. We did not quantify this quantity, as our main focus was on stability, or in other words, on reducing the systematic drifts to minimum. This latter question is more directly related to the quantifications in Ref. [5], where the key interest was in the number of true attractors, and not on the speed of convergence when starting at intermediate states. Relating our results, in a rate model network, to those of Ref. [5] (in a binary network) is interesting, as mentioned in the Discussion.
>
> Please see also our first response to reviewer m6Fs.

---

> > ### Comment · Reviewer_fsQk · 2023-08-13
> > **post rebuttal**
> >
> > Thank you for the replies and clarifications.
> >
> > Regarding the approach of ensuring a number of patterns are fixed points, and the relation to continuous attractors, this reference might also be relevant:
> > Darshan, R., & Rivkind, A. (2022). Learning to represent continuous variables in heterogeneous neural networks. Cell Reports, 39(1).
> >
> > I'm maintaining my score.

---

> > > ### Author Response · Authors · 2023-08-14
> > >
> > > Thanks for the suggestion, we agree that this reference is relevant in this context and will include it in the final version.

---

### Official Review · Reviewer_NH89 · 2023-07-05

**Soundness:** 3 good
**Presentation:** 2 fair
**Contribution:** 2 fair
**Rating:** 5
**Confidence:** 4

**Summary:**

A new method is proposed to allow the simultaneous embedding of multiple attractors in an RNN through minimization of the energy function corresponding to the dynamics.
Two different methods to achieve this are considered, the first one based on the linearized energy function and the second on  constrained optimization. The second method optimizes the weights for a flat energy landscape with the constraint that there is no change in the center of mass of the bumps.


**Strengths:**

The aim to flatten the energy landscape of an attractor network with multiple attractors embedded in it is novel.
The main aim and the methods are clearly described.
The discussed problem of detrimental inference is definitely very important for theoretical neuroscience.

**Weaknesses:**


There is a lack of assurance that the modified network actually has the equalized energy function maintains the bumps as minima of the energy landscape.


The procedure also doesn't necessarily contribute to a flat energy landscape, it just ensures that all the evaluated states have the same energy value.


A more thorough way to enforce flattening energy functions for RNNs is described in: Noorman, Marcella, et al. "Accurate angular integration with only a handful of neurons." bioRxiv (2022): 2022-05.


The time to implement the algorithm is also really long 72 hours for $L=60$ and does not seems to be practical.

The reason to use the bump score is not fully justified.

**Questions:**

174: To analyse the modified network, one can use the linearized network or Lyapunov exponents to identify stability. Try to find the equilibria of the modified network and assess the stability of these points. Do the equilibrium states have a Lyapunov exponent close to 0? (This is expected from the ring attractor structure.)


CANNs can implement path integration. Is it possible to do that reliably in the proposed network for the different embedded maps?


However, the proposed idealized bump method to do this does not guarantee that the acquired energy landscape is truly flat. Can you plot the energy landscape for values in between bump positions?

SI 111: Do you mean that the function $f$ is defined \emph{in terms of} the weighted averaged position of the rates? The definition that follows is different.

132-133: Could you further explain how these steps follow from each other? The gradient of $I^{k,l}$ in S24 is not dependent on the constraint, why couldn't the derivative of it with respect to $M_{ij}$  change in another  direction?

190: Could you further assess the stability of the network by introducing small displacements in the activity?
What are the Lyapunov exponents of the network? Do they correspond to a continuous attractor?

194: Could a different definition of the location lead to a different conclusion from the analysis?
The actual attractor states are not the idealized attractors any more necessarily.
Therefore, the mean square change defined like this might be biased.

**Limitations:**


They are adequately discussed.

---

> ### Author Rebuttal · Authors · 2023-08-07
>
> >There is ... landscape.
>
> Existence of the Lyapunov function rigorously guarantees that network dynamics will converge to stationary states which are local minima of the energy (lines 96-97). We have precisely characterized these minima, and carefully verified our numerical scheme by checking that the energy function is locally quadratic near these minima. Nevertheless, we agree that there is value also in checking independently that convergence takes place, as additional verification of the numerical scheme. Please see Supp Figs. 1 and 2b.
>
> >The procedure ... value.
>
> This is true only for the first scheme in which we evaluate the energy of idealized bump states. Next, we find the true minima of the energy, constrained on the center of mass of the bump thereby precisely evaluating the energy landscape as a function of bump position. Our optimization scheme then aims to flatten this landscape, and we succeed in doing so as shown in Fig. 4c and Supp Fig. 3b.
>
> >A more ... (2022):
>
> Noorman et al aimed to generate a *single* continuous ring attractor with a small number of neurons. Their approach, like ours, is based on the flattening of a Lyapunov energy landscape. Their work treats a very specific form of connectivity with cosine weights that enables the derivation of some results analytically, but is difficult to generalize to other forms of connectivity. In this sense, our approach is more general. In fact, we successfully used our approach to address the problem of Noorman et al using other forms of initial connectivity. In the *multi-map* problem addressed in our submission, the random permutations yield highly irregular weight profiles, and the problem of flattening the landscape is considerably more difficult.
>
> >The time ... practical.
>
> Our focus is on asking a basic theoretical question: whether it is possible to embed multiple flat continuous attractors in a single network? We did not put effort into optimizing our code or running it on state-of-the-art hardware and it is likely that there are more efficient ways to do so. For example, much of the ~72 hours mentioned above is due to our lack of an automated pipeline for gathering each iteration results and generating the next one. The key value of our work is in showing that the flattening is possible, and not in the efficiency of the algorithm used to obtain this result.
>
> >The reason ... justified.
>
> In the literature on Hopfield based models, the retrieval of a memory is obtained by computing the intuitive overlap measure between the memory pattern and the network state. This is equivalent to the computation of position using our bump score. Previous works that studied multiple embedded maps have used this measure as well [32,23,1]. For completeness, we also used the population vector to measure the position of localized activity. Both measures have shown a dramatic improvement in the network drift (Supp Fig. 4). We now clarify this choice in the SM.
>
> >174: ... structure.)
>
> Since rate networks with symmetric weights are guaranteed to settle on stationary steady states, it’s sufficient to examine the Jacobian of the dynamics to assess stability. If the attractor is continuous, one expects to observe an eigenvalue close to zero. We didn’t check this explicitly, but the slow dynamics over a continuum of near-steady states and the near-flatness of the energy function (Figs. 4c and 5a,d,e, and Supp Figs. 1 and 3b) implies so.
>
> >CANNs ... maps?
>
> It has been argued that path integration within the hippocampus would be difficult, because the synaptic connectivity required to do so would need to be tailored for each embedded map. Thus, it was suggested that path integration occurs elsewhere - perhaps in the entorhinal cortex. This idea was recently examined in Ref. [1], demonstrating that path integration can reliably be implemented despite the interference between the maps. The flattening procedure discussed in our work is expected to substantially improve the accuracy of this computation.
>
> >However, ... positions?
>
> As the reviewer notes, Fig. 2a,b for 10 maps only shows an approximation to the energy landscape, because the idealized bump states are not true minima of the energy. We later evaluate the precise energy landscape by finding states that minimize the Lyapunov function under a constraint on their center of mass. This is done at a dense set of positions (Fig. 4a and Supp Fig. 3b), with single neuron resolution. Even the classical ring attractor composed of N neurons is not precisely continuous, but has N precisely steady states. This is an issue of practical significance only when the number of neurons is small (Noorman et al).
>
> >SI 111: ... different.
>
> Yes, fixed.
>
> >132-133: ... direction?
>
> I^kl in Eq. S24 is defined as the state that minimizes the energy under the constraint. This is where the dependence on the constraint is coming from. Any modification of I^kl due to changes in M must remain in the subspace in which the constraint is obeyed. We modified the SM to clarify this point. In addition, note that we numerically verified that the second term in Eq. S24 vanishes (Supp Fig. 2b).
>
> >190: ... attractor?
>
> The steady states that we identify are minima of the energy under the constraint on the center of mass, and therefore the manifold of states that we identify is stable. The stability of the approximate attractor is evident in Fig. 5a,d,e. See above response regarding the Lyapunov exponent.
>
> >194: ... biased.
>
> This is true for our idealized bump scheme (Fig. 3c), where the mean squared change (MSC) is compared between the converged and the initialized idealized bump states. However, in Fig. 5 we quantify the stability of states that are true minima of the energy under a constraint on the position of the bump: we start from these deformed bump states and follow their dynamics (SM lines 84-85). Since the MSC is compared between deformed converged states of consecutive iterations there is no bias in relation to idealized bumps.

---

> > ### Comment · Reviewer_NH89 · 2023-08-15
> >
> > I thank the authors for their replies and clarifications.
> >
> > About the flatness of the energy landscape
> > When do you consider the number of neurons to be small? I agree that truly continuous attractors are only achieved when the number of neurons is infinite, but that would imply that anything smaller than that is a "small" number of neurons?
> > Is there a different sense in which you understand "small" for the "practical significance" you mention?
> >
> > There seems to be a misunderstanding about my reference to the Lyapunov exponent (https://en.wikipedia.org/wiki/Lyapunov_exponent). I understand Fig 5 to contain information about the Lyapunov energy, however, I think it would benefit the analysis and would make the claims stronger about stability. To me it is unclear how the measure of stability in the paper relates to a resolution that is below the spatial resolution of one neuron (which should be considered for a continuous attractor).
> >
> > I maintain my score.

---

> > > ### Author Response · Authors · 2023-08-15
> > >
> > > >About the flatness of the energy landscape When do you consider the number of neurons to be small? I agree that truly continuous attractors are only achieved when the number of neurons is infinite, but that would imply that anything smaller than that is a "small" number of neurons? Is there a different sense in which you understand "small" for the "practical significance" you mention?
> > >
> > > There are several ways in which we understand "small" from a practical standpoint. The first is simple: if there are, say, a thousand neurons participating in a ring attractor with a single map, the graininess of the energy landscape is 2pi/1000. In other words, a stable representation can be maintained at the resolution of a full circle divided by the number of neurons. Whether or not this is small in a practical sense depends on the context, but it is no coincidence that the question addressed in the work of Noorman et al was brought up only recently, with the discovery of a ring attractor with a few dozens of neurons. In CA3, one can estimate that tens of thousands of neurons participate in the representation of each square meter map. Note, in addition, that the granularity arising from the finite number of neurons is small, within the parameters that we work with, compared to the tuning curve of each neuron.
> > >
> > >
> > > Second, the drifts that we address in our work, which arise from the embedding of multiple maps, are much larger than the single neuron resolution - see Figure 3d. The single neuron resolution seems to us like a natural scale for our attempts to flatten the energy landscape, since this is the graininess which is expected without further manipulations even if only a single map is embedded in the connectivity.
> > >
> > >
> > > In principle, however, we could have attempted to equalize the energy over a set of positions which is more dense than the single neuron resolution. Our methodology can be applied to achieve this goal even in the single map case - in similarity to the results of Noorman et al, but using an approach which is in some ways more general. We were able to do so for various forms of the single-map connectivity, and were even able to obtain an exquisitely stable representation of positions over a continuum of angles in a network consisting of only three neurons. We did not include these results in our submission because the focus of the present work is on the consequences arising from the embedding of multiple maps.
> > >
> > >
> > > Third, in a realistic neural network one expects to have two types of noise: frozen noise in the connectivity, which in our context arises from the embedding of multiple maps, and dynamic noise that arises from the fact that neural activity is dynamically stochastic. Dynamic noise causes diffusive random motion in the position of the bump, which accumulates over time. The characteristic magnitude of the diffusive motion over a short time interval Dt scales as the square root of Dt/N, where N is number of neurons, whereas the graininess arising from the discrete number of neurons scales as one over N. In addition, the energy barriers reduce with N.. Consequently, random diffusive motion dominates the motion over short time scales, and the systematic drifts associated with the discrete number of neurons are completely washed out by the random diffusive motion when N is sufficiently large. With biophysically reasonable parameters, and when N is in the order of several hundred or more, the statistics of random motion in a single ring attractor is practically indistinguishable from the statistics predicted analytically using the continuum limit, while completely ignoring the graininess that arises from the finite number of neurons (see, e.g., https://doi.org/10.1073/pnas.1117386109).
> > >
> > > >There seems to be a misunderstanding about my reference to the Lyapunov exponent (https://en.wikipedia.org/wiki/Lyapunov_exponent). I understand Fig 5 to contain information about the Lyapunov energy, however, I think it would benefit the analysis and would make the claims stronger about stability. To me it is unclear how the measure of stability in the paper relates to a resolution that is below the spatial resolution of one neuron (which should be considered for a continuous attractor).
> > >
> > > Lyapunov exponents are primarily used to identify chaotic dynamics. In our case, there is no question that the system has stable attractors, and it cannot have a positive Lyapunov exponent. You are correct, that a system with a semi-stable continuous attractor should have a vanishing Lyapunov exponent. What we pointed out in our previous response is that in a system with stable attractors, a somewhat simpler signature for the continuity is that the Jacobian has a vanishing eigenvalue. This has indeed been used previously in the literature on attractor neural networks as a way to quantify the continuity of the attractor, and we could add such an analysis to the final version.

---

### Official Review · Reviewer_vC81 · 2023-07-10

**Soundness:** 3 good
**Presentation:** 3 good
**Contribution:** 2 fair
**Rating:** 4
**Confidence:** 4

**Summary:**

This work tackles the problem of interference between continuous attractors when they are held in a single RNN.  The authors adopted a Lyapunov function as a depiction of energy of the network and tried to flatten the energy landscape of attractors by adding a modulation term to the original connection matrix.  The modulation term was derived by two methods respectively: a first-order approximation of the energy function and a constrained iterative gradient-based method.  They showed that by constraining bumps at their initial position, the gradient descent achieved a better result.

**Strengths:**

The work achieved the goal of encoding multiple continuous attractors into a single recurrent network.

**Weaknesses:**

1. The recent experimental data actually showed that in the remapping of cognitive maps in hippocampus, place cells encoding different maps actually have little overlap, i.e., the hippocampus recruits different groups of neurons to form different continuous attractors. In other words, the interference between multiple continuous attractors is only a mathematical problem, not a biological problem. This limits the contribution of this study to neuroscience.
2. There are several issues in this study whose biological plausibility are not justified, the energy function (note the real neuronal connections are not symmetry), the modification of synapse strengths based on the attractors the network has stored, the gradient-based learning method. Overall, the insight of this study to neuroscience is rather limited.
3. Some important references are missed, such as the work of Misha Tsodyks et al. on stroring multiple continuous attractors (PLoS Computational Biology?)


**Questions:**

1. Since the learning method is used to determine M, why the authors use W=\sum_i J_i+M and learn the modification term, why not learn W from the first space?

**Limitations:**

The limitations are discussed.

---

> ### Author Rebuttal · Authors · 2023-08-07
>
> >The recent experimental data actually showed that in the remapping of cognitive maps in hippocampus, place cells encoding different maps actually have little overlap, i.e., the hippocampus recruits different groups of neurons to form different continuous attractors. In other words, the interference between multiple continuous attractors is only a mathematical problem, not a biological problem. This limits the contribution of this study to neuroscience.
>
> Remapping of CA3 (and CA1) place fields in distinct environments (“global remapping”) has been documented in a large number of studies (see, for example, the following two review articles:
> https://doi.org/10.1016/j.tins.2008.06.008, https://doi.org/10.3389/fnbeh.2017.00253).
>
> Typically, in a given small environment, only a subset of CA3 place cells are active. Therefore, some cells will be active only in one environment and not in another. However, in any two environments, there is a significant fraction of cells that express firing fields under both conditions. The hallmark of global hippocampal remapping is that the spatial relationship between these fields is unrelated in the different environments.
>
> As an example, see:
> https://doi.org/10.1038/nature05601
> Figure 1 shows firing rate maps of a few CA3 cells under global remapping, and Supporting Figure 2 (panels A and B) shows rate maps of many additional cells.
>
> We are therefore not sure why the reviewer comments that the hippocampus recruits distinct, non-overlapping groups of neurons to represent different environments. We may have misunderstood the comment of the reviewer, and therefore we kindly ask that the reviewer will reexamine this claim.
>
> >There are several issues in this study whose biological plausibility are not justified, the energy function (note the real neuronal connections are not symmetry), the modification of synapse strengths based on the attractors the network has stored, the gradient-based learning method. Overall, the insight of this study to neuroscience is rather limited.
>
> Progress in the theoretical understanding of the brain has benefited from various approaches and styles of research: among these approaches, those based on mathematical abstraction have often proved to be useful, because this facilitates the identification of general principles. For example, the Hopfield model of associative memory was formulated in terms of binary units, with symmetric connectivity, and using many other gross idealizations of the biological reality. Despite these choices – and, in fact, perhaps because of them, the model had tremendous influence on the thinking about short-term and long-term memory, and the relation between neural network organization and function. Numerous later studies expanded on this model. Some examined how the same principles can be realized with more biological realism, while others used the model in unexpected contexts not foreseen originally.
>
> Likewise, most models of head-direction, grid-cell, and hippocampal networks to date assume symmetric connectivity, as in our study. This is not meant to suggest that the question of non-symmetric connectivity is unimportant: however, by providing a proof or principle that multiple continuous attractors can be embedded without compromising the stability (as presumed previously) we provide a new insight – even if we do so at this stage only for symmetric connectivity. It is likely that our work will motivate future studies, in which some of the limitations mentioned by the reviewer (and discussed in the section of our manuscript on limitations) will be confronted.
>
> Furthermore, the value of rigorous mathematical theory is often realized in unexpected contexts. The ring attractor model was initially proposed for orientation selectivity on V1, yet it has motivated the formulation of theoretical models of head direction cells, and has been recently extremely valuable in understanding the physiology and structure of the fly’s central complex, where surprisingly tight correspondence has been observed between network organization (and dynamics) and the seemingly idealized mathematical theories that were developed in previous studies for head-direction cells.
>
> We therefore believe that our study provides a significant development in the theory of attractor networks in the brain, and that it fits well into the scope of NeurIPS.
>
> >Some important references are missed, such as the work of Misha Tsodyks et al. on stroring multiple continuous attractors (PLoS Computational Biology?)
>
> We were not sure which paper was meant here, and wonder whether it is this one:
> https://doi.org/10.1371/journal.pcbi.1000869.
> This article proposes a network architecture in which two or more maps are embedded in a single network, in the case where the patterns of activity associated with the maps are correlated. The paper does not address the problem of achieving precise stability, yet it is thematically related to our work in a broad sense because of the study of simultaneously embedded maps. We will be happy to cite it, and will be grateful if the reviewer could provide us with other concrete suggestions for additions to the reference list.
>
> >Since the learning method is used to determine M, why the authors use W=\sum_i J_i+M and learn the modification term, why not learn W from the first space?
>
> We first started from the established naive form of connectivity J, which was identified and used in previous studies for embedding multiple attractors. This initial connectivity can be viewed as arising from a Hebbian form of learning, as discussed in previous studies. We implemented a *perturbative* approach in which we started from this known approximate solution to the problem, where the energy landscape (as we show) is not precisely flat. The perturbative approach is key to our methodology (please see lines 137-152), and the weight corrections that are required to rescue the continuity of the attractor are indeed small (Fig. 4b).

---

> > ### Comment · Reviewer_vC81 · 2023-08-13
> >
> > Thanks for the authors insightful response. Most of my concerns are resolved.
> > Regarding to the remapping of place cells, the authors could see this work (https://www.pnas.org/doi/10.1073/pnas.1421056111), whose results exhibited that the overlap is minimal between the recruited place cells in different environments.

---

> > > ### Author Response · Authors · 2023-08-14
> > >
> > > Thanks for pointing to the manuscript of Alme et al (PNAS, 2014). This work is helpful because it quantifies the overlap more systematically than in previous studies. The recordings made by Alme et al were carried out mostly in novel environments and only for 15 minutes, and therefore it is possible that spatial maps were not yet fully established in the CA3 network (see also Leutgeb et al, 2004). Nevertheless, the recordings are informative on a qualitative level, and the conclusion is that overlap is small, but not minimal. In the context of our work the overlap observed by Alme et al is extensive and consequential, as we explain below.
> > >
> > > The interpretation reached by Alme et al is that most CA3 place cells are active in ~7% of the environments and a smaller subset of cells, estimated in the paper to be 10%-20% of the total population, are active in a much larger fraction of the environments. Furthermore, the picture emerging from the paper is not of a strict division of cells into clusters, each representing a distinct map. The results are consistent with a picture in which the participation of each cell in any given environment is determined randomly and  independently of the other cells.
> > >
> > > In terms of the whole population, the above estimates suggest that the participation ratio (defined as the fraction of cells that are active in any given map) is somewhere between 10% and 20%. A similar conclusion can be reached by asking what is the probability that a cell will be active in a map B, given that it is active in map A (and is therefore already classified as a place cell). This information can be extracted from the histogram shown in Fig. 4A: the cells that had a field in one map or more participated, on average, in the representation of 2.75 rooms out of 11. Excluding the environment used to classify the cell as a place cell, the participation ratio is approximately no less than 1.75/10 = 0.175 ( (2.75-1)/(11-1)).
> > >
> > > With a participation ratio of ~0.15, neurons expressing a bump state in one environment are expected to receive numerous synaptic inputs resulting from weights associated with any other map. These will add up in proportion to the number of embedded spatial maps. Furthermore, many studies indicate that the hippocampus encodes memories other than spatial maps, which are expected to contribute as well to the frozen noise. Overall, the effect of frozen noise associated with the embedding of multiple memories will depend on p, on L (the number of spatial maps, and other associated memories) and on N, the number of neurons participating in each map. Importantly, the flatness of each attractor state is inevitably compromised in the naive embedding scheme. The magnitude of the effect must increase in proportion to the number of embedded memories, and therefore it must become large for sufficiently large L.
> > >
> > > For computational simplicity we implemented in our simulations a network architecture in which all cells participate in all maps. However, it would be straightforward to adapt the architecture to one with a participation ratio smaller than 1. We agree that it will be interesting in a follow-up study to examine (numerically and using analytical tools) the interplay between the participation ratio p, the number of embedded maps L, and the number of neurons participating in each map N. Note that small p implies low interference, which is beneficial since it reduces the frozen noise, but when keeping the total number of CA3 neurons fixed, it also implies a reduction in the number of neurons participating in each map, and in each bump state. This is expected to reduce the resilience of each attractor to the frozen noise.
> > >
> > > We thank the reviewer for raising these questions. We will add a paragraph in the Discussion of the final version, in which we will discuss the fact that in reality, neurons in CA3 participate only in a subset of maps, and how this could be addressed in future studies.

---

### Official Review · Reviewer_m6Fs · 2023-07-24

**Soundness:** 4 excellent
**Presentation:** 4 excellent
**Contribution:** 4 excellent
**Rating:** 8
**Confidence:** 3

**Summary:**

The authors present a new technique for embedding multiple attractor manifolds into RNNs. To do so, they first randomly choose a number of attractor manifolds, embed these into an RNN, and then make weight adjustments to smooth out the interference created by multiple manifolds. The authors propose two strategies for weight adjustment, which consider first- and then second-order interference effects. These adjustments are shown to iteratively improve several intuitive metrics.

**Strengths:**

This paper addresses and interesting and well-defined problem. This is not my area of expertise, but the results seem quite general to understanding RNN function, and thus significant.

The paper is well written and the results clearly presented. The stability metrics are intuitive.

The presented solutions are strikingly effective!

**Weaknesses:**

While the results are very strong, the paper only explores a single task (embedding 1D ring attractors). Do the results hold when moving to, say, two dimensions? How does the number of neurons in the network interact with task complexity?

**Questions:**

the phrase "quenched noise" is used a lot - can this be defined more clearly upon introduction? My idea of "quenched" is equivalent to "reduced", which does not seem to align with how this phrase is being used.

L15: typo: adjustment -> adjustments
L151: typo: "emfbedded"

Fig 1b top: only 9 red lines - need to expand axis limits?

Fig 2b top: it is interesting that modification - while flattening the energy landscape - also raises the overall energy of the system. It would be instructive if the authors could offer some intuition for why this is the case. Also, in Fig 4a it appears that the energy after second-order modification is actually less than the single map case - will this always be the case? How should I think about this result?

Figs 3-5: what are error bars? The captions say mean+/-SEM, but what are these statistics computed over? Are there multiple network instantiations for each value of L?

**Limitations:**

One limitation that is already pointed out by the authors is that their approach is not a biologically plausible learning algorithm; however, I agree that this proof-of-principle is a solid first step and that investigating other mechanisms for weight updates is an interesting direction for future work.

---

> ### Author Rebuttal · Authors · 2023-08-07
>
> >While the results are very strong ... task complexity?
>
> In order to reduce the computational cost, we explored our schemes in 1D, but it is straightforward to extend our approach and implement it in 2D. Conceptually, we do not expect a qualitative difference when doing so. One notable difference between 1D and 2D environments is that the number of neurons required to achieve a good approximation to a continuous attractor, even for a single map, scales in proportion to the area in 2D, as opposed to length in 1D. But for a given number of neurons, there is no substantial difference between the two cases in terms of the complexity of the problem: the number of equations scales as N*L, and the number of parameters (synaptic weights) scales as N^2. Since the random permutations are completely unrelated to the spatial organization of the firing fields, quenched (frozen) noise is expected to behave similarly in the two cases.
>
> Please see also our last response to reviewer fsQk.
>
> >the phrase "quenched noise" ... being used.
>
> The term 'quenched noise' is used in statistical physics to describe frozen disorder in many-particle systems (as opposed to noise and disorder arising from dynamic fluctuations). We defined this term in our context in the introduction (line 54): it describes a hard-wired (and thus also referred to as frozen) noise in the connectivity which is independent of time. This is in contrast to dynamical noise that affects instantaneous neural firing rates (see also Refs. [23,5]).
>
> We will add the following clarification in the final version:
> “Unlike stochastic dynamical noise which is expressed in instantaneous neural firing rates, time-independent quenched noise is hard-wired in the connectivity – it breaks the symmetry between the continuum of neural representations.”
>
> >L15: typo: adjustment -> adjustments L151: typo: "emfbedded"
>
> Both fixed.
>
> >Fig 1b top: only 9 red lines - need to expand axis limits?
>
> Yes! Fixed, thanks.
>
> >Fig 2b top: it is interesting ... why this is the case.
>
> The (flat) modified energy landscapes in Fig. 2b (top and bottom, orange traces) are not the true energy landscapes, because the energy is evaluated using idealized bumps and these are not true steady states of the multi-map network. These precisely flat energy traces are shown as sanity check, to demonstrate that the added weight modifications indeed produce the expected outcome of a flat energy landscape, when re-evaluating the energy landscape using idealized bumps. For this reason, the comparison between the absolute energy values of the pre-modified (blue) and modified (orange) landscapes in Fig. 2b is not fully justified.
>
> Subsequently in the manuscript, we performed a precise evaluation of the energy landscape, by replacing the idealized bumps with steady states obtained through the constrained gradient optimization scheme. The appropriate figure to look at is therefore Supplementary Fig. 3b, where we show the precise energy landscape across multiple iterations. Here it is evident that the overall (mean) energy does increase, as suggested by the reviewer.
>
> In our scheme, we do not seek to reach a prescribed value of the energy in each iteration, but only to equalize the energy across states at a value which is determined implicitly in the optimization scheme. This introduces a bit more freedom in the choice of the correction weights than if targeting for a prescribed energy. We suspect that the increase of the energy in our scheme is specific to the choice of the single-map connectivity that we worked with, and that other choices of this connectivity could elicit a decrease of the energy during optimization. We did not check this systematically, however, since uniform shifts of the energy landscape are inconsequential for the stability and dynamics of the bump states.
>
> >Also, in Fig 4a ... about this result?
>
> We did not show how the energy depends on the number of embedded maps. We realize now that Fig. 2a might be confusing in this respect: throughout the manuscript, we uniformly shifted the energy by a constant in order to allow for the landscapes to be shown on the same plot for a single map and for 10 maps (Fig. 2a). This constant was selected such that the mean energy of idealized bump states across all states and maps (without any weight modification) is zero: thus, the mean of the blue trace in the bottom panel of Fig. 2b is zero (as well as single map blue traces, Fig. 2a). Therefore, comparison of absolute mean energies in Fig. 2a between single map and 10 maps traces is not meaningful. This is now explicitly explained in the final version.
>
> We can, however, compare the absolute mean energy value for landscapes with a varying number of maps before shifting and centering them around 0: when doing so, we find that as more maps are embedded in the connectivity, the absolute mean energy of the landscape decreases. To understand why this happens one should look at the expression for the energy (Eq. 1). The network parameters (Supplementary Material) were chosen such that the sum of rows/columns of the basic connectivity matrix is a constant value. Therefore, each additional map embedded in the connectivity will contribute, when averaging over a random permutation, a constant shift to the energy through the third term of Eq. 1. This constant is negative for our choice of the single-map connectivity matrix.
>
> In summary, there is a systematic shift of the mean energy with addition of new maps that depends on the specific choice of the single-map connectivity matrix, and occurs even without any weight modifications. The weight modifications introduce additional shifts in the energy, but these are fairly subtle compared to the pre-modified dependence of the mean energy on the number of maps.
>
> >Figs 3-5: ... value of L?
>
> Yes, please see Supplementary Material lines 138-141. We added in the final version a reference to the Supplementary Material in the captions of Figs 3-5 (after mentioning SEM).

---

> > ### Comment · Reviewer_m6Fs · 2023-08-10
> >
> > Thank you for the thorough response.
> >
> > Regarding 1D vs 2D environements, this is likely a common question that will come up among readers. I'd suggest at least adding something to this effect, perhaps in the Discussion?
> > > One notable difference between 1D and 2D environments is that the number of neurons required to achieve a good approximation to a continuous attractor, even for a single map, scales in proportion to the area in 2D, as opposed to length in 1D. But for a given number of neurons, there is no substantial difference between the two cases in terms of the complexity of the problem: the number of equations scales as N*L, and the number of parameters (synaptic weights) scales as N^2. Since the random permutations are completely unrelated to the spatial organization of the firing fields, quenched (frozen) noise is expected to behave similarly in the two cases.
> >
> > While I don't think it's strictly necessary to include a 2D example in the manuscript, I think the impact (and at least perceived generality) of this work would increase with said example (even a small one, even in the supplementary).
> >
> > My other concerns/confusions have been adequately addressed.

---

> > > ### Author Response · Authors · 2023-08-14
> > >
> > > Thanks for the suggestion, we will mention and address this (2D) in the Discussion of the final version.

---

### Author Rebuttal · Authors · 2023-08-07

We thank the reviewers for their careful reading of our submission and for their insightful comments. Please see our point-by-point responses to each review. We will highly appreciate any additional comments or requests for clarification that may arise in response to our answers to the questions.

---

### Decision · Program_Chairs · 2023-09-21

**Decision:**

Accept (poster)

**Comment:**

This paper presents a new method for embedding multiple attractor manifolds into recurrent neural networks. Naive embedding of such manifolds lead to interference between them. The method described here fine tunes the synaptic weights in a way to reduce this interference. This is an important contribution that will be helpful in building mechanistic models of brain function.